# Genetically encoded discovery of perfluoroaryl macrocycles that bind to albumin and exhibit extended circulation in vivo

Jeffrey Y. K. Wong [1,8], Arunika I. Ekanayake[1,8], Serhii Kharchenko[1], Steven E. Kirberger[2], Ryan Qiu [1], Payam Kelich [3], Susmita Sarkar[1], Jiaqian Li[2], Kleinberg X. Fernandez[1], Edgar R. Alvizo-Paez[1], Jiayuan Miao [4], Shiva Kalhor-Monfared[1], J. Dwyer John[5], Hongsuk Kang[6], Hwanho Choi[6], John M. Nuss[5], John C. Vederas [1], Yu-Shan Lin [4], Matthew S. Macauley [1,7], Lela Vukovic [3], William C. K. Pomerantz [2] & Ratmir Derda [1] ✉

Peptide-based therapeutics have gained attention as promising therapeutic modalities, however, their prevalent drawback is poor circulation half-life in vivo. In this paper, we report the selection of albumin-binding macrocyclic peptides from genetically encoded libraries of peptides modified by perfluoroaryl-cysteine $S_N$Ar chemistry, with decafluoro-diphenylsulfone (**DFS**). Testing of the binding of the selected peptides to albumin identified SICRFFC as the lead sequence. We replaced **DFS** with isosteric penta-fluorophenyl sulfide (**PFS**) and the **PFS**-SICRFFCGG exhibited $K_D = 4-6\,\mu M$ towards human serum albumin. When injected in mice, the concentration of the **PFS**-SICRFFCGG in plasma was indistinguishable from the reference peptide, SA-21. More importantly, a conjugate of **PFS**-SICRFFCGG and peptide apelin-17 analogue ($N_3$-$PEG_6$-NMe17A2) showed retention in circulation similar to SA-21; in contrast, apelin-17 analogue was cleared from the circulation after 2 min. The **PFS**-SICRFFC is the smallest known peptide macrocycle with a significant affinity for human albumin and substantial in vivo circulation half-life. It is a productive starting point for future development of compact macrocycles with extended half-life in vivo.

There are around 80 peptide drugs on the global market; more than 150 peptides are in clinical development, and another 400–600 peptides are undergoing preclinical studies[1]. In contrast to typical small molecule drugs, the large surface area of peptides allows them to interact with

expanded binding interfaces commonly found in protein–protein interactions, protein–carbohydrate, and protein–DNA interactions. These proteins, classified as "undruggable targets," have been difficult to target using conventional small molecule therapeutics but many of

[1]Department of Chemistry, University of Alberta, Edmonton, AB T6G 2G2, Canada. [2]Department of Chemistry, University of Minnesota, Minneapolis, MN 55455, USA. [3]Department of Chemistry and Biochemistry, University of Texas at El Paso, El Paso, TX 79968, USA. [4]Department of Chemistry, Tufts University, Medford, MA 02155, USA. [5]Ferring Research Institute, San Diego, CA 92121, USA. [6]Quantum Intelligence Corp., 31F, One IFC, 10 Gukjegeumyung-ro, Yeongdeungpo-gu-Seoul, Republic of Korea. [7]Department of Medical Microbiology and Immunology, University of Alberta, Edmonton, AB T6G 2E1, Canada. [8]These authors contributed equally: Jeffrey Y. K. Wong, Arunika I. Ekanayake. ✉e-mail: ratmir@ualberta.ca

them have been addressed by peptide, proteins, or antibody therapeutics. Peptides are the smallest among the latter three modalities−2 to 10 kDa for peptides versus 150 kDa for full-sized antibodies−and they possess distinct pharmacokinetic (PK) properties. For example, biodistribution of peptides and small proteins inside tumours and other non-vascularized tissues is improved compared to full-size antibodies. Several clinical candidates (TH1902, TH1904, BT5528, BT8009, BT1718, MMP-14) capitalize on such improved bio-distribution[2–4]. Unlike antibodies, which remain in circulation for 1–3 weeks due to the association with the neonatal Fc-receptor (FcRn) on the surface of endothelial cells[5–7], peptide therapeutics clear within minutes to hours from plasma by renal filtration. Fast clearance is a beneficial property in several therapeutic applications, such as imaging (e.g., "tumour paint"), radionuclide delivery (e.g., Lurathera™)[8], and in administration of short-acting peptide hormones. However, for the more widespread adaptation of peptide modalities in diverse therapeutic applications, it is desired to tune the circulation lifetime of peptides from minutes to hours.

Only a few peptides exhibit a naturally extended circulation lifetime: a therapeutically relevant example is a natural venom, 39-residue peptide 'exendin 4', with low renal clearance in humans (5–7 h)[9]. This peptide gave rise to the FDA-approved drug exenatide for treating type 2 diabetes[10,11]. Despite favourable circulation half-life, modified derivatives of exenatide−liraglutide, albiglutide, dulaglutide, lixisenatide, and semaglutide[1]−have been developed to further tune circulation half-life and other PK properties. The majority of peptides and small proteins have to be modified as well to increase their circulation time. Such modification could be divided into several classes: Class 1: increase in size via covalent linkage to polyethylene glycol (PEG)[12,13], polyglycerol[14], and other synthetic macromolecules. Interestingly, steric hindrance by these size-increasing moieties also protects against proteolytic degradation[15–17]; Class 2: increase in size via controlled oligomerization[18]; Class 3: covalent linking to long-living serum protein (e.g., FDA-approved drugs albiglutide and dulaglutide, exenatide that conjugated to albumin and the IfG4 Fc domain)[19–21] and Class 4: incorporation of moieties that bind non-covalently to serum proteins such as albumin[22–25], immunoglobulin[26,27], FcRn[28], transthyretin[29], and transferrin[30,31]. An important example in the last class is the lipidation of peptides to allow interaction with serum albumin. Lipidation has been one of the most successful strategies to prolong the half-life of peptides and small proteins such as insulin, giving rise to FDA-approved drugs such as Levemir®, Tresiba®, Victoza®, Saxenda®, and Ozempic® with extended serum half-life[32,33]. The improved properties of these and many other drugs stemming from their association with albumin mandate investigation of albumin as a carrier for therapeutic applications.

Albumin is the most abundant protein in plasma, with an average concentration of 600 μM and an average half-life of 19 days in humans[34]. The main mechanism leading to the long half-life of albumin and antibody are similar: both proteins interact with FcRns on the surface of endothelial cells[6,7]. This binding results in transient endocytosis of these proteins, and as a result, they are frequently sequestered from circulation and protected from clearance. At physiological pH, the binding affinity between albumin and FcRn is low; however, the interaction under acidic conditions in the endosome is strong to avoid lysosomal degradations and recycling of albumin to the extracellular space[5]. Albumin is a versatile carrier of essential fatty acids and diverse small organic molecules[34]. Among all the long-circulating serum proteins, albumin is considered to be one of the most important targets because of its ability to interact with hydrophobic small molecule drugs and enhance their pharmacokinetic properties. The recurrent therapeutic success of rationally lipidated peptides and proteins[35] fuels interest in rational development of small molecules as well as non-lipidated proteins and peptides that bind to albumin.

Many FDA-approved small molecule drugs have an intrinsic affinity for human serum albumin (HSA). Targeted development of small molecules with high affinity for HSA has been a topic of research over the last 15 years (see recent review, ref. 25). Anti-HSA antibodies, nanobodies[36], DARPins[37], and other protein domains have been also developed. Such proteins can be fused to therapeutic proteins of interest to extend their in vivo circulation. Similarly, short peptides that bind to HSA could be used in tandem with therapeutic peptide or protein sequences to dial in predictable half-life for such therapeutics. Such short albumin-binding peptides could empower development of many future therapeutic peptides because they could be built into *any* genetically encoded peptide library (e.g., displayed on phage, RNA, and other platforms) to give rise to billion-scale libraries with predictable in vivo half-life. However, short HSA-binding peptides are scarce. A 31-mer peptide DX-236 (Ac-AEGTGDFWFC-DRIAWYPQHLCEFLDPEGGGK-NH$_2$) with a binding affinity of 1.9 μM was identified by Dyax Corp., and used to purify albumin (Fig. 1A)[38]. A 21-mer peptide SA-21 (Ac-RLIEDICLPRWGCLWEDD-NH$_2$) with a binding affinity of -0.5 μM to HSA was identified at Genentech (Fig. 1B)[39] and subsequently conjugated to ligands for urokinase-type plasminogen activator[22,40], Fab antibody fragments[41,42] and small proteins[43] to prolong their circulation half-lives. Heinis and co-workers developed a short heptapeptide modified by fluorescein isothiocyanide (FITC) and palmitic acid (FITC-EYEYK$_{palm}$ESE-NH$_2$) with a binding affinity of 39 nM to HSA (Fig. 1C)[23], and the presence of both lipid moiety and fluorescein was critical for the binding of this peptide. This FITC-lipopeptide was fused with two different bicyclic peptides to boost the half-lives from minutes to hours[23]. Success of DX-236, SA-21, and FITC-lipopeptide, and other examples from the literature demonstrated the possibility of using HSA as a target for genetically-encoded selection to identify HSA-binding peptides with extended circulation half-life. Despite availability of generic lipid-based albumin binders, antibodies and protein-based binders to albumin, and small molecule-based human-albumin binders, there remains an interest in developing other classes to develop small peptide-based albumin-binding ligands with lower molecular weight[23,39]. In this work, we employ genetically encoded phage-displayed libraries of chemically modified macrocycles to develop albumin-binding mini scaffolds. To hone on the shortest possible peptide sequences, we employed a phage-displayed libraries SXCX$_n$C, $n = 3$–5 modified with decafluorodiphenyl sulfone (DFS)[44,45] where X is any amino acid except for cysteine (Fig. 1D). We hypothesized that a perfluoroaromatic linchpin might serve as a useful pharmacophore and it might be recognized by one of the binding sites of HSA similar to the binding of fatty acid; however, as we observed in the NMR studies and late-stage pharmacokinetic evaluation of albumin-binding macrocycles, the primary function of perfluoro-linchpin is to constrain the discovered peptide macrocycle in a productive albumin-binding conformation. While perfluoroaromatic linchpin alone does not equip a random peptide with albumin-binding properties, the change in the shape of perfluoroaromatic linchpin is detrimental to the albumin-binding properties and in vivo circulation.

## Results
### Selection of albumin binders
We devised and conducted three discovery campaigns that used different library architecture and selection strategies. In the first discovery campaign, we modified the phage libraries of structure SXCX$_{4–5}$C with DFS following a previously published protocol and confirmed that 85% of the phage library is modified to yield octafluoro-diphenylsulfone-cross-linked macrocycles (**OFS**-SXCX$_{4–5}$C-phage) (Fig. 2A, Supplementary Fig. 1A)[44]. We performed three rounds of phage selection using HSA coated to the surface of 96-well polystyrene plates as bait. In parallel, we screened the same library on polystyrene wells coated with Protein A (negative control) to distinguish specific HSA-binding sequences from poly-specific protein binding sequences

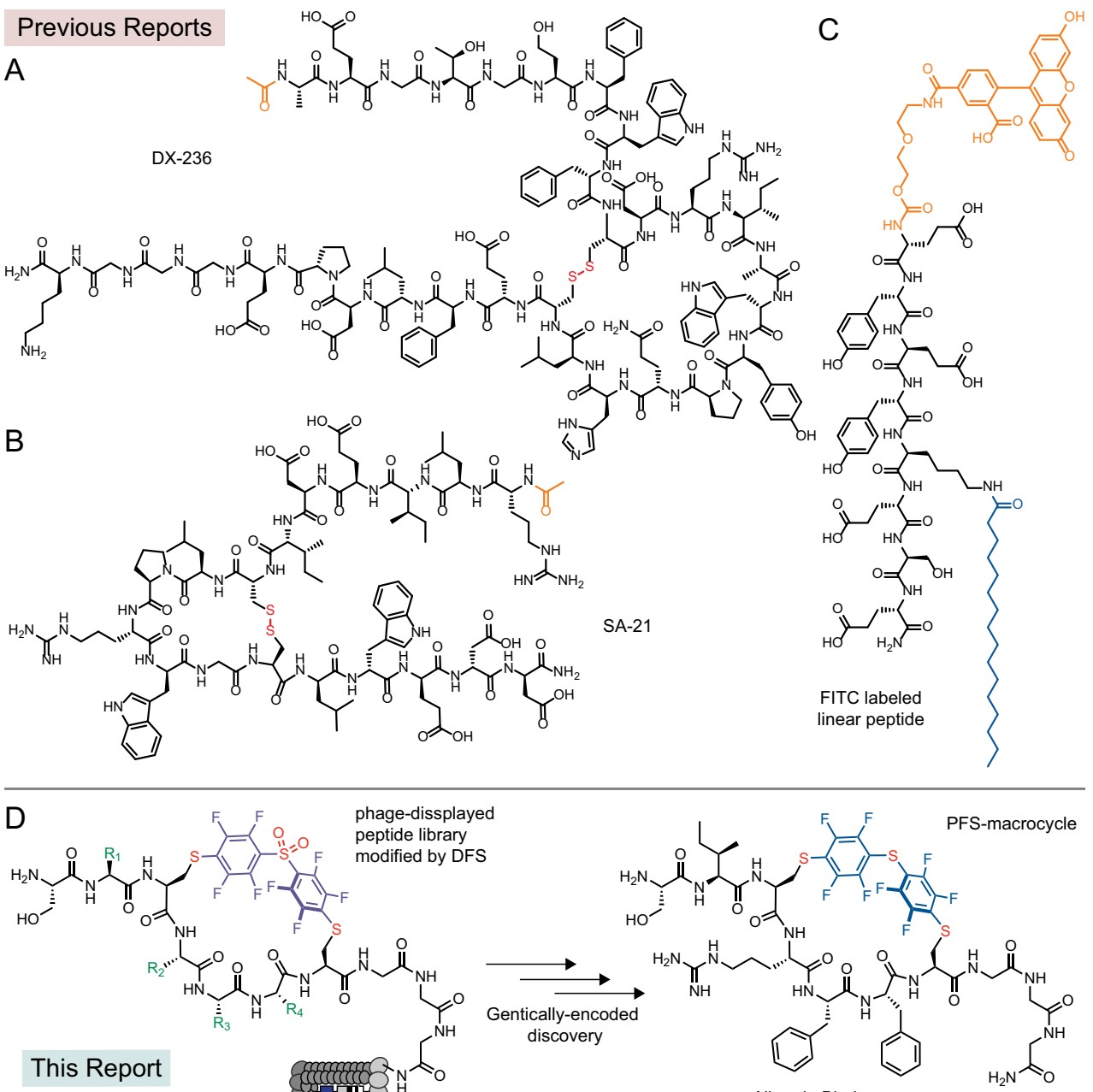

**Fig. 1 | Albumin binding peptides.** Previous reports of **A** macrocyclic peptide: DX-236[38], **B** macrocyclic peptide: SA-21[39], **C** a linear peptide: FITC-EYEYK$_{palm}$ESE-NH$_2$[23]. **D** This report describes a chemically modified phage-displayed library for discovery of a small macrocyclic albumin binder.

(Supplementary Fig. 1A). In round 3, the recovery of the OFS-macrocycle library against HSA was 17-fold lower than the recovery of the unmodified library, indicating that the OFS linchpin contributes selection; however, the increase in round-to-round phage recovery was only modest. Despite depletion on protein A, the enriched population bound equally to HSA and protein A (Supplementary Fig. 1D). To mitigate these problems, the second discovery campaign increased the stringency and altered the immobilization of HSA between 96-well plate in rounds 1 and 3 and biotinylated HSA immobilized onto streptavidin beads in round 2 (Supplementary Fig. 2A). Such changes to selection stringency led to 200-fold increase in recovery in round 3 when compared to rounds 1 and 2. The recovery of the unmodified library panned against HSA was insignificant (Supplementary Fig. 2D). The binding of the **OFS**-macrocycle phage library recovered from round 3 to Protein A, ConA, and Casein was 2, 14, and 300-fold lower,

respectively, compared to recovery on HSA-coated wells (Supplementary Fig. 4). These observations suggested that (i) specific albumin-binding sequences had been selected and (ii) the binding of these sequences to albumin required the presence of an **OFS** linchpin (Supplementary Fig. 4). Differential enrichment (DE) analysis of the next-generation sequencing (NGS) of all test and control experiments (Supplementary Table 1) identified families of peptide macrocycles that had statistically significantly higher ($p < 0.05$) enrichment in binding to HSA when compared to binding to unrelated protein (Supplementary Figs. 1 and 2). DE analysis of the first selection yielded three consensus motifs: STCHDITC (**1**), STCHYIGC (**2**) and STCHANC (**3**) (Supplementary Fig. 1E) whereas the second selection campaign yielded consensus motif: STCHTIYC (**4**) (Supplementary Fig. 2E). Although the original libraries were designed as SXCX$_n$C where $n = 4$ and 5, and the SXCX$_3$C sequences exist in such libraries only as rare

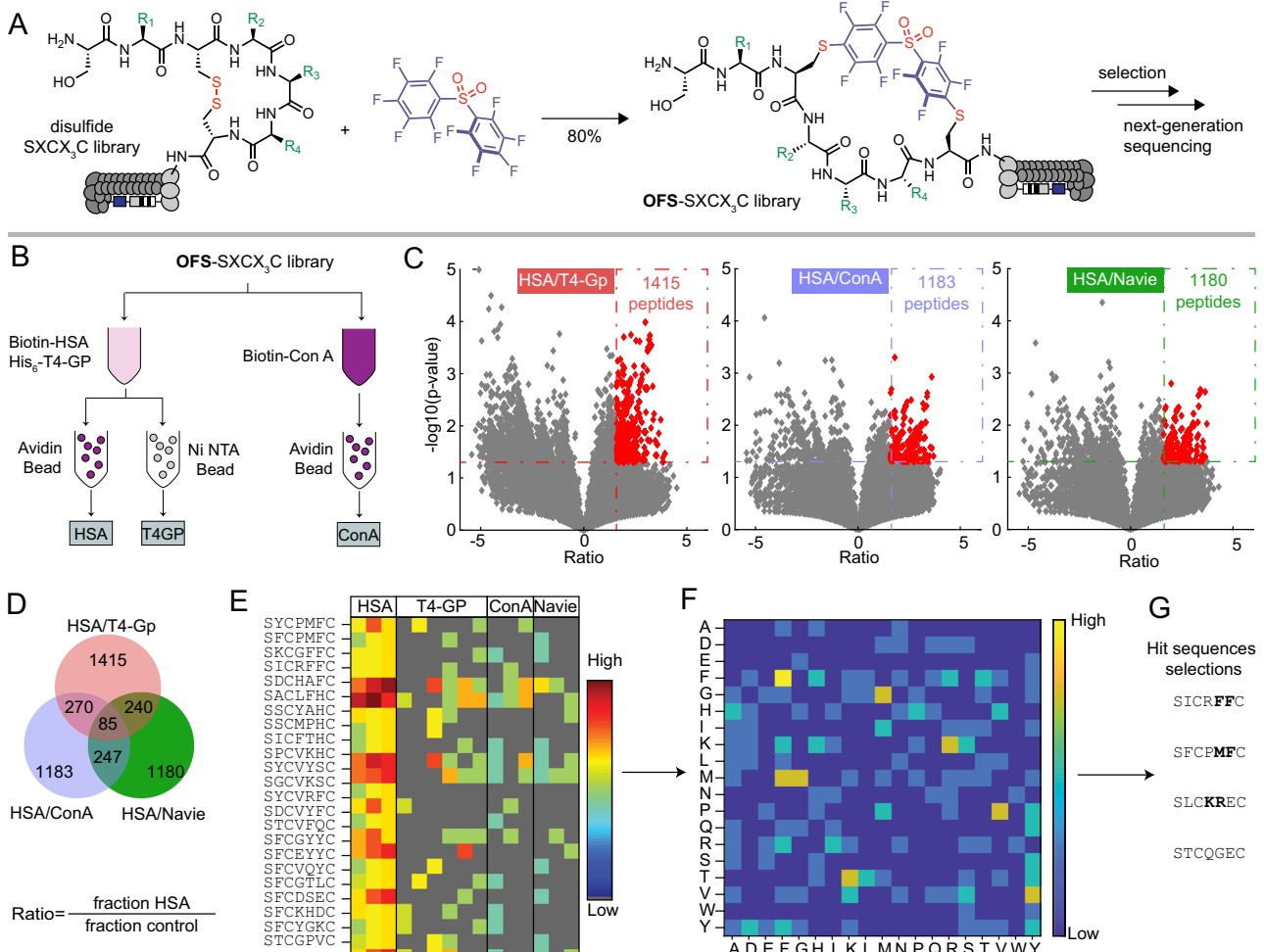

**Fig. 2 | Panning with chemically modified phage libraries. A** Modification of phage-displayed SXCX$_3$C disulfide library by **DFS** to yield **OFS**-SXCX$_3$C library. **B** The **OFS**-SXCX$_3$C library panned against a mixture of biotinylated HSA and His-tag expressed T4-GP in solution containing unlabelled milk proteins. Targets were captured separately with avidin beads and Ni-NTA beads affinity beads. In the negative control, **OFS**-SXCX$_3$C library was panned against biotinylated ConA and captured with avidin beads. **C** Volcano plots and **D** Venn diagram visualizing the sequences from the **OFS**-SXCX$_3$C phage-displayed library that were significantly enriched in the HSA screen compared to the naive library or selection against T4-GP, ConA. **E** A heat map display of the top 25 of 85 hits sequences from differential enrichment results. **F** Dipeptide motif analysis of all 85 hits. **G** Selected sequences for chemical synthesis of macrocycles for validation.

deletions[46], we still observed the enrichment of these rare sequences in the selection. Intrigued by the preference for smaller macrocycles, we devised a third selection campaign that employed only SXCX$_3$C libraries modified with **DFS** (Fig. 2A). We used previously reported biotin capture[44] to confirm that **OFS** linchpin can modify SXCX$_3$C library to 88% conversion (Supplementary Fig. 3).

The SXCX$_3$C library has been used in our previous publications successfully using only one round of sequencing[46–48]. It was sufficiently small (160,000 variants) to be fully covered by NGS and copy number of each member of the naive library was sufficiently large to afford quantification of the selection by NGS and DE analysis. To maximize the stringency of one-round-selection, we replaced immobilized albumin with soluble albumin as a screening bait. Soluble bait minimized preference of multivalent phage for multivalent display of albumin on the surface and avoided any detrimental change in albumin structure resulting from its immobilization on polystyrene surface. To mimic the complex serum environment, the panning mixture contained unlabelled milk proteins mixed with biotinylated HSA (Bio-HSA), His-tag fusion T4-PG protein (His$_6$-T4-PG). After addition of phage, HSA and T4-PG were captured from this mixture with streptavidin or Ni-NTA affinity beads, respectively. Control selection employed the same mixture with biotinylated ConA (Bio-ConA)

instead of Bio-HSA (Fig. 2B). The captured phage DNA was liberated from beads by treatment with hexane, amplified by PCR (Supplementary Fig. 6) and analysed by NGS (Fig. 2B). A DE analyses of selections against Bio-HSA, His$_6$-T4-GP and Bio-ConA identified significantly enriched ($p < 0.05$, >3-fold) 85 hit sequences (Fig. 2C–E, Supplementary Fig. 4); their pairwise amino acid clustering identified FF and other motifs (Fig. 2F). From the combined analyses, we nominated sequences SICRFFC (**5**), SFCPMFC (**6**) and SLCKREC (**7**) as hits and STCQGEC (**8**) as a negative control for chemical synthesis and further validation (Fig. 2G). In summary, three selection strategies against HSA yielded divergent binding motifs. Such divergence of selection campaigns is not surprising because protein immobilization, depletion and amplification strategies were different among these strategies selection. To compare the sequences from these screens, we synthesized a series of hits and tested their binding to albumin using a battery of biochemical assays.

## Validation of albumin binders

We observed non-specific reactivity of **OFS**-macrocyclic peptides with thiol nucleophiles such as glutathione (GSH) over several hours in basic pH (Supplementary Fig. 7). Replacing **DFS** with a less reactive pentafluorophenyl sulfide abolished the undesired reactivity: The

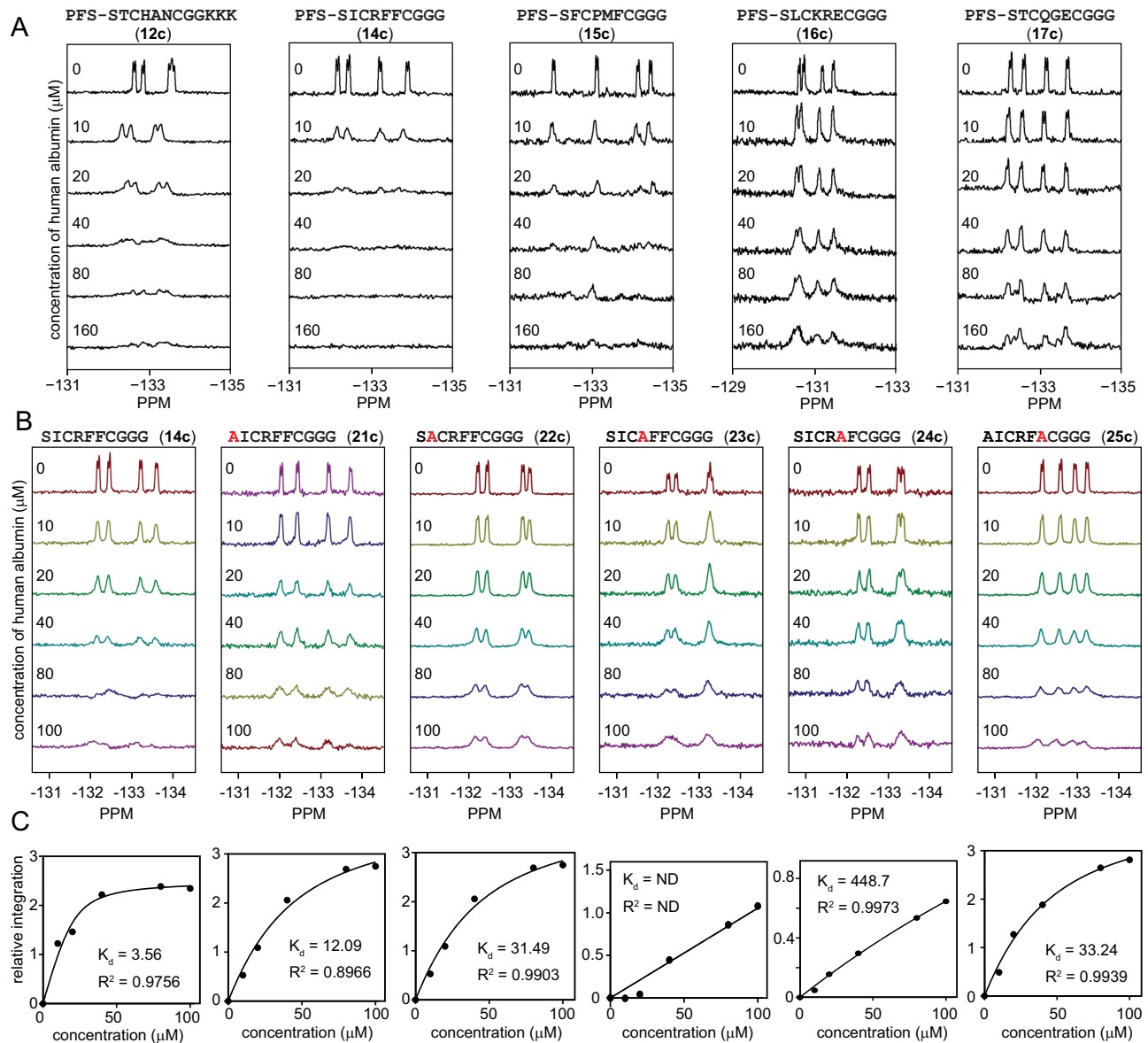

**Fig. 3 | ¹⁹F NMR measurement of macrocycle-albumin interactions. A** ¹⁹F NMR binding assay for macrocycles **12c** (**PFS**-STCHANCGKKK), **14c** (**PFS**-SICRFFCGGG), **15c** (**PFS**-SFCPMFCGGG), **16c** (**PFS**-SLCKRECGGG), and **17c** (**PFS**-STCQGECGGG) at 20 μM against varying concentrations of HSA. **B** ¹⁹F NMR signals and **C** extrapolated binding curves for **14c** (**PFS**-SICRFFCGGG) and five alanine mutants (**21c, 22c, 23c, 24c, 25c**) of this macrocycle.

resulting perfluorophenylsulfide (**PFS**)-macrocycles were unreactive to 2-mercaptoethanol over 3 weeks and unreactive towards free thiol on HSA (Supplementary Fig. 8). Molecular dynamics simulation suggested the **OFS**-macrocycles and the **PFS**-macrocycles exhibit similar ground-state conformational landscapes (Supplementary Fig. 9). Poor solubility of perfluoro-aryl cross-linked macrocycles **1–8** (STCHDITC, STCHYIGC, STCHANC, STCHTIYC, SICRFFC, SFCPMFC, SLCKREC, STCQGEC) made it difficult to evaluate their interaction with albumin in aqueous buffers. To increase their solubility, we synthesized these sequences with GGKKK or GGG tag at the C-terminus. The C-terminal tags provided sufficient solubility properties for downstream analyses. Some sequences were synthesized with both tags to verify that these tags did not affect binding to HSA.

The unique fluorine handle in perfluoro-aryl-modified peptides made it possible to determine their binding to HSA using ¹⁹F NMR (Fig. 3A). In a typical experiment, we maintained peptide concentration at 50 μM and varied the concentration of HSA from 10 to 100 μM (**4**). The broadening of and disappearance of ¹⁹F signals that correspond to

fluoroaromatic groups indicated the binding of the peptide to HSA (Fig. 3A and Supplementary Fig. 10A). It was not always possible to fit a definitive $K_D$ value to the binding response due to the complex binding behaviour and quality of the NMR signal. However, in an albumin titration series, it was straightforward to obtain a qualitative estimate such as the albumin concentration necessary to suppress 50% of the initial fluorine signal. After evaluation of eight sequences found in all discovery campaigns (Fig. 3A, Supplementary Figs. 10–11), we nominated **PFS**-SICRFFCGGG (**14c**) as the lead for further evaluation and **PFS**-STCQGECGGG (**17c**) as the negative control in further investigation.

A substantial change in measured $K_D$ values resulting from subtle amino-acid changes in **PFS**-macrocycles highlighted that the binding is not driven by the perfluoroaromatic linker alone. The peptide sequence plays a major role. Alanine scan of **14c** further reinforced this observation (Fig. 3B, C). We observed that R4A or F5A changes in **14c** ablated the binding between the perfluoro-macrocycles and HSA. In contrast, the S1A and I2A changes resulted in smaller changes in $K_D$ of

 

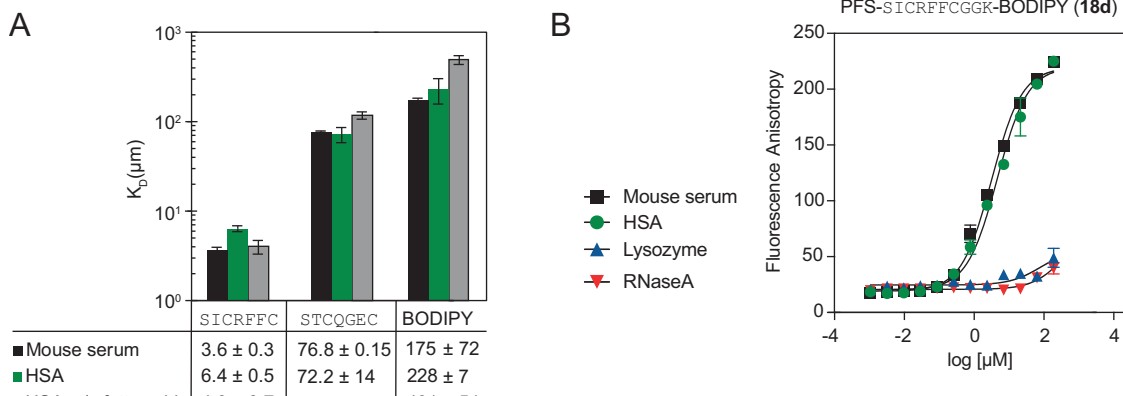

**Fig. 4 | Binding between selected macrocycles and HSA measured using fluorescent polarization. A** FP assay measured the $K_D$ of macrocycles **18d** and **19d** against mouse serum, HSA, and fatty acid-free HSA ($n = 3$). **B** The FP assay for

BODIPY labelled **18d** titrated against mouse serum (black), HSA (green), lysozyme (red), and RNAse A (blue) ($n = 2$). Bars represent mean values ± SD.

3- and 10-fold, respectively. The alanine scans indicate that variation to any of the four discovered amino acids in SXCXXXC sequence negatively affects the binding. Even serine, a constant residue in selection, becomes modestly important in albumin recognition as well. We titrated **14c** and **17c** against rat serum albumin and observed similar binding to rat and human albumin (Supplementary Fig. 12). We attempted to confirm the binding affinity of these sequences by isothermal titration calorimetry (ITC) using SA-21 as a control[39]; however, a complex multi-site binding behaviour for all peptides obscured the accurate evaluation of binding affinity by ITC (Supplementary Figs. 13–15), and this multi-step binding to albumin in ITC experiments has been observed in previous reports[49]. The [19]F NMR assay, thus, was critically enabling for validation and ranking of the albumin binding leads.

A fluorescence polarization binding assay (FP) served as a complementary assay to confirm the binding affinities of the macrocycles labelled with the fluorophore BODIPY at the C- (**18d**) or N-terminus (**14e**). In a typical experiment, we used **PFS**-SICRFFCGGK (**18d**) or **PFS**-STCQGECGGK (**19d**) at 1 μM concentration and titrated HSA from 0.1 to 100 μM. The dose-response curve could fit a single-state binding model with a binding affinity of $K_D$ = 4–6 μM for **18d** and at least 100 times weaker affinity for **PFS**-STCQGECGGK (**19d**) (Fig. **4**A and Supplementary Fig. 16). BODIPY bound weakly to HSA with >300 μM binding affinity (Fig. **4**A and Supplementary Fig. 16). The FP assay like the previous [19]F NMR assay could also measure binding to other proteins or even complex mixtures (serum). A titration of the mouse serum (Supplementary Fig. 16C) yielded a similar binding profile to that observed in binding to pure albumin (Fig. **4**A). Replacing HSA with lysozyme and RNAse, the assay detected no binding response, confirming that **18d** binding was specific to HSA (Fig. **4**B). Switching the location of the fluorescent probe from the N-terminus to C-terminus did not significantly change the affinity of **14c** ($K_D$ = 4–6 μM, Supplementary Fig. 17). The switching from **DFS** (**18f**) to **PFS** (**18d**) exhibited no difference in FP experiment (Supplementary Fig. 18A), but minor differences were observed by NMR (Supplementary Fig. 18B). The results from FP were in the same order of magnitude as the semi-qualitative estimates acquired for BODIPY-free peptides by the [19]F NMR binding assay, indicating that the presence of a fluorophore did not significantly increase the binding (Supplementary Fig. 19). Heinis and co-workers observed that fluorophores could dramatically increase binding affinity of peptides for albumin, and removing the fluorophore can be detrimental to the binding of the albumin ligand[23]. To exclude this possibility, we tested **PFS**-SICRFFCGG macrocycles with and without BODIPY in [19]F NMR binding assay and observed similar binding affinity (Supplementary Fig. 19).

## Elucidation of the binding pocket for perfluoro-macrocycles

To identify the binding location of **PFS**-SICRFFCGGG (**14c**), we attempted to co-crystalize **14c** with HSA but were not successful. As an alternative, we attempted to map the binding pocket of **14c** by testing whether carbamazepine, diclofenac, or ibuprofen inhibit the binding of **14c** (Supplementary Fig. 20). As none of these drugs influenced the binding of **14c**, we concluded that primary binding pocket for **14c** does not coincide with the primary binding pockets of carbamazepine, diclofenac, or ibuprofen. To identify the plausible binding location of **14c**, we performed a series of docking calculations. Nine distinct sites on HSA bind to fatty acids[50], some of which also accommodate ibuprofen and diclofenac[51,52] (Supplementary Fig. 21, Fig. **5**A). Docking of **14c** to these nine sites identified binding site 1, which is close to Hemin-binding site, as the most favourable (Fig. **5**A). The docking score for site 1 averaged over all the docking calculations across five distinct HSA structures (PDB 1E7E, 1E7F, 1E7G, 1E7H, 1E7I) was −8.95 ± 1.0 kcal/mol whereas the scores for the next sites were −6.6 ± 1.1, −6.2 ± 0.7, and −6.0 ± 1.2 kcal/mol (Supplementary Figs. 22 and 23). Observed docking preference away from known ibuprofen and diclofenac sites corroborated the experimental observation (Supplementary Fig. 20). As site 1 is hemin-binding site, a desired experimental validation of this prediction would be inhibition of HSA:**14a** interaction by hemin. We attempted to measure **14c**:HSA interactions in the presence of hemin, but the results were inconclusive due to strong association of hemin and **14c** in [19]F NMR experiments. We note many drugs have primary and secondary (lower affinity) binding sites on HSA surface and further discussion on how multi-site binding affects prioritization of site 1 is provided Supplementary information section 28 "Docking Calculations".

To further evaluate prioritization of albumin site 1 as a plausible binding site for **14c**, we performed NVT (constant temperature, constant volume) and NPT (constant temperature, constant pressure) simulations of **14c** in site 1 of HSA solvated with explicit TIP3P water molecules and calculated binding free energy using steered molecule dynamics (SMD) and umbrella sampling techniques. We then calculated the potential of mean force (PMF) of the unbinding process in biased MD simulations with a harmonic potential whose interaction center is located at a specific distance between the binding pocket and the center of mass of **14c**. The ΔG was about −7.0 kcal/mol for pocket 1, while the ΔG for the other pockets was less than −4.2 kcal/mol. The PMF corroborated that pocket 1 has significantly stronger binding energy than the others (Fig. **5**A, Supplementary Fig. 26). Furthermore, the calculated ΔΔG of free energy perturbation calculations (FEP) for **14c** and five Ala mutants of **14c** were in alignment with $K_D$ measured for the same alanine mutants in the [19]F NMR assay (Fig. **3**B, C). The biggest loss of function for R4A mutant highlighted the importance of Arginine

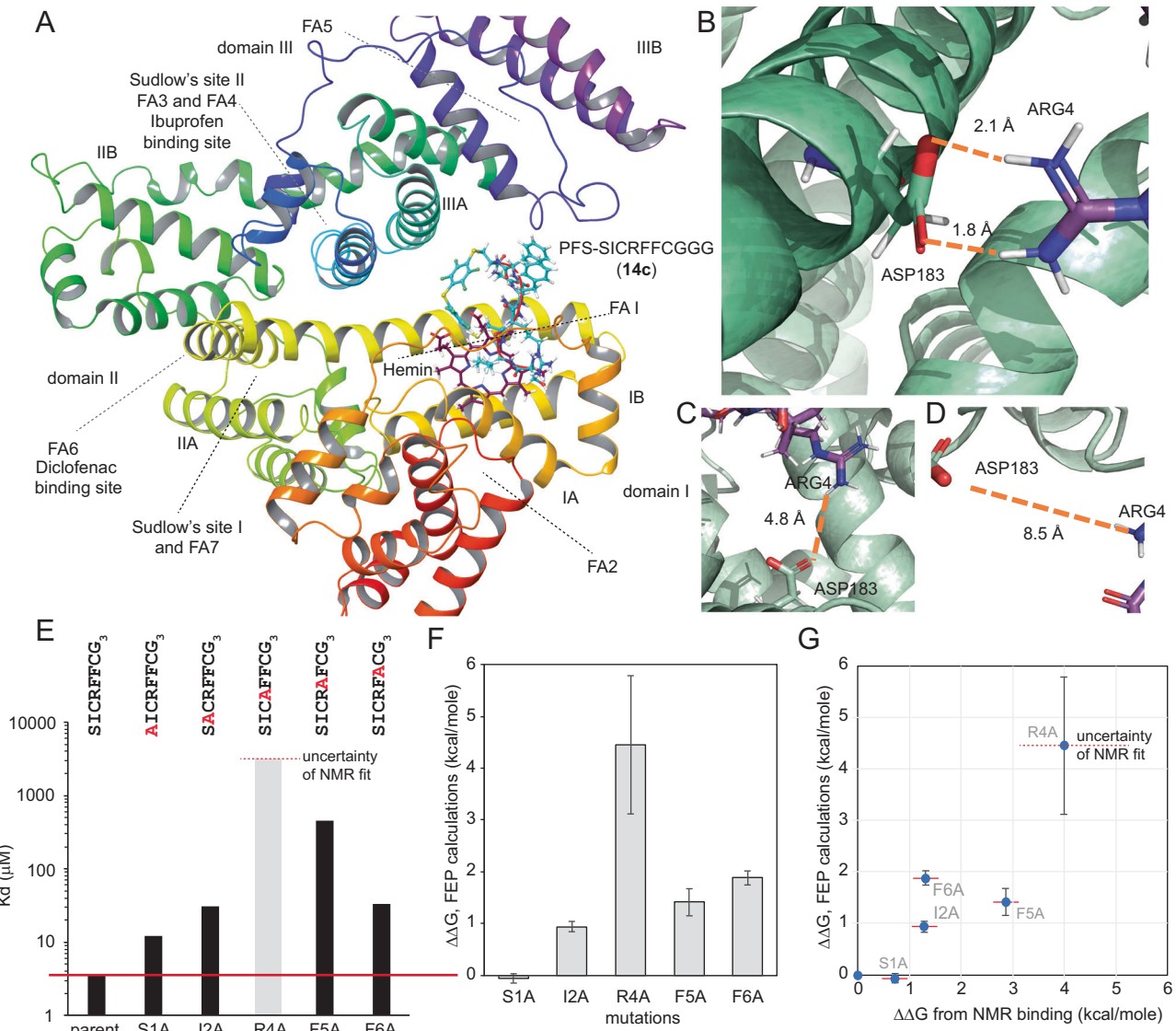

**Fig. 5 | Docking predictions of HSA-14c complex. A** Macrocycle **14c** shows the lowest binding energy when binding to pocket 1 near the Hemin-binding site of HSA; three helical domains (DI, DII, DIII), the subdomains (A and B) and binding sites for ibuprofen and diclofenac are mapped on the same HSA structure. **B** SMD and umbrella sampling trajectories in site 1 revealed that ARG4 of the **14c** forms interaction for albumin binding (Fig. 5E–G); in contrast, both NMR and stable salt bridges with ASP183 of the albumin whereas this salt bridge is absent in other binding modes (**C, D**). **E** $K_D$ values measured by $^{19}$F NMR. **F** ΔΔG for alanine point mutations calculated by free energy perturbation (FEP). **G** Correlation of experimental ΔΔG calculated from $K_D$ and ΔΔG from FEP calculation for the alanine mutants.

FEP agreed on a relatively minor role of N-terminal Ser and minor loss for S1A mutant.

In summary, while we failed to co-crystalize **14c** with HSA, the combined findings from FP and NMR, drug inhibition, docking/MD, and aligned performance of Ala-mutants in FEP and NMR studies offer a useful guide for optimization of **14c** and its use in delivery. For example, both the binding pose in site 1, and NMR/FP studies suggest that both C- and N-termini of the **14c** might be accessible as a plausible location for the attachments of payloads to **14c**. We followed up on these suggestions in pharmacokinetic studies.

## Circulation lifetime of 14c derivatives in mice

We evaluated the half-life of **14c**, its derivatives, and SA-21 peptide as a positive control in mice by administrating a mixture of peptides into mice's tail vein followed by HPLC/MS analysis of plasma samples at 2 and 60 min. The concentration of SA-21 in plasma at 2 and 60 min decreased subtly from $4 \times 10^4$ to $3 \times 10^4$ ng/mL whereas concentration of

macrocycle **14c** in plasma was 3 and $1.5 \times 10^4$ ng/mL at the same time points (Fig. 6B). In contrast, four derivatives of **14c** were excreted at 2 min or were not detectable at 60 min. The fast-clearing analogues were linear SICRFFCGGG with the cysteines alkylated by iodoacetamide (**14l**), the alanine mutants **PFS**-SIC**A**FFCGGG (**23c**) and **PFS**-S**A**CRF FCGGG (**21c**), and the control **PFS**-ST**CQGE**CGGG sequence (**17c**); poor plasma retention of the last three analogues was in agreement with their significantly decreased association with albumin in the NMR studies (Fig. 3). Replacing **PFS** by hexafluorobenzene (**HFB, 14j**) and deca-fluorobiphenyl (**DFB, 14k**) did not ablate retention in plasma completely (Fig. 6A); however, concentration of **HFB**- and **DFB**-modified macrocycles after 60 min were factor of 10 lower when compared to **PFS**-modified **14c** parent. These observations reinforced the important structural features of **14c**: both the amino acids sequence of **14c** and specific conformational constraint of this sequence imposed by **PFS** linker are critical for even a short-term retention in circulation.

In the same short-term circulation study, we examined the several N- and C-terminal derivatives of **14c** to identify the permissive location

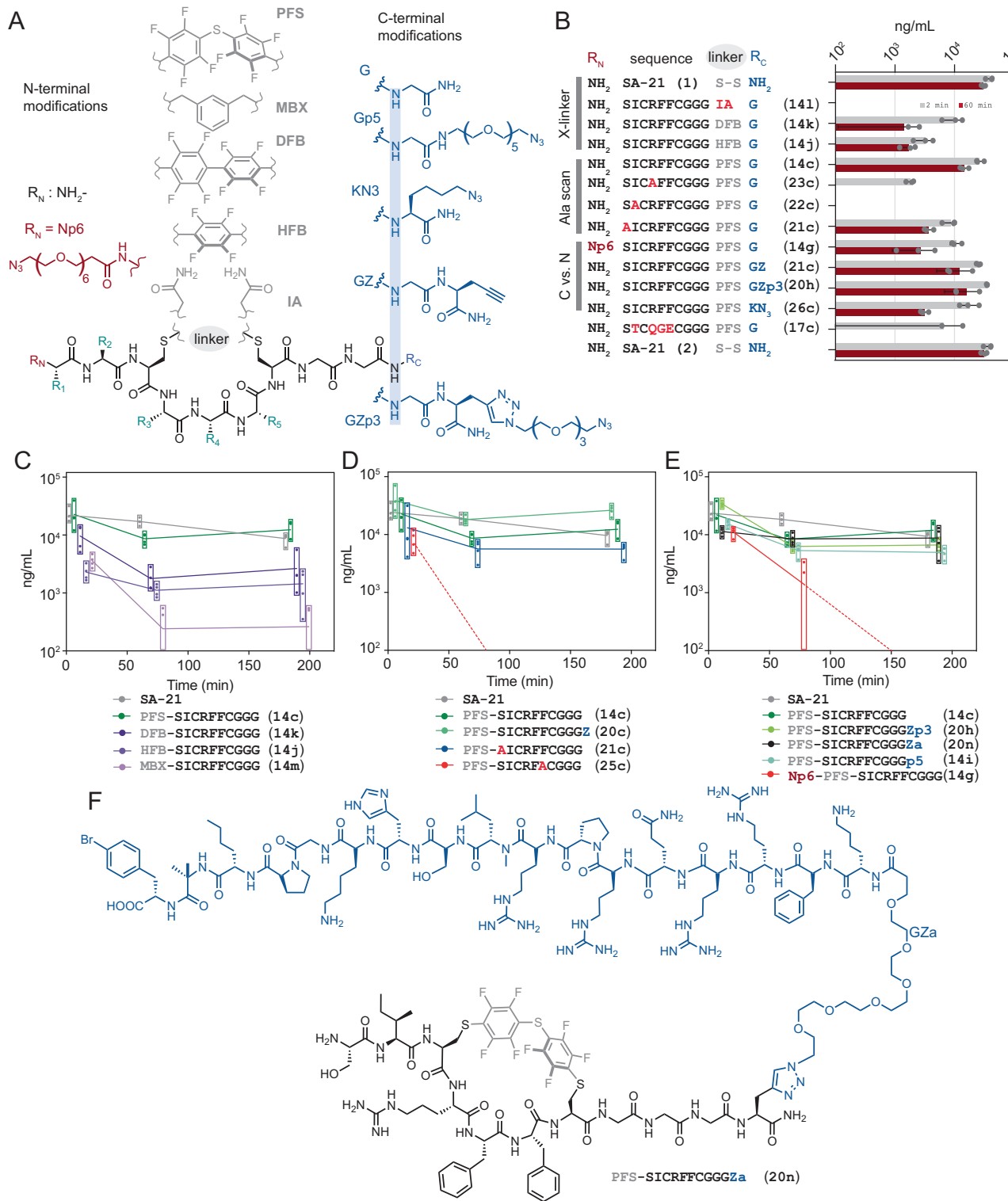

**Fig. 6 | Pharmacokinetic studies of SICRFFCGGG macrocycle in mice serum.**
**A** Lead sequence SICRFFC modified with N- and C-terminal extensions and different linchpins. **B** A mixture of compounds, including SICRFFC modifications, alanine scans, together with SA-21 injected into mice and monitored for their retention up to 1 h (n = 3 mice for each peptide). **C** N- and C-terminal extensions on **PFS**-SICRFFC macrocycle for payload attachments monitored for circulation time. n = 4 mice for each peptide. **D** Monitoring the effects of alanine mutants and C-terminal propargyl glycine on retention time. n = 4 mice for each peptide. **E** Effects of different linchpins on circulation time. n = 4 mice for each peptide. **F** 14c conjugated to therapeutically relevant payload, apelin-17 analogue (N₃-PEG₆-NMe17A2). Bars and boxes represent mean values ± SD.

in **14c** molecule for the immobilization of the cargo. The two N-terminal derivatives of the **14c** were macrocycle **PFS**-A̲ICRFFCGGG (**21c**) with Ser to Ala replacement on N-terminus and macrocycle **14g** synthesized from **14c** by acylation of N-terminal Ser with azido-PEG₆-

carboxylic acid. Both **21c** and exhibited a factor of 10 decrease in concentration after 60 min when compared to **14c** parent. In NMR studies, modifications at the N-terminus led to a factor of 3 decrease in albumin binding capacity (Fig. 3) and the combined observations

suggest that N-terminus is not an optimal location for modification of **14c**. We then examined the retention of three C-terminal derivatives: **PFS**-SICRFFCGGK(N3) (**26c**) where K(N3) denotes a γ-azidolysine, **PFS**-SICRFFCGGGZ (**20c**), where Z indicated propargyl glycine and **20h** synthesized by Cu-catalysed 1,3-dipolar cycloaddition (CuAAC) of **20c** and di-azido-PEG$_3$ linker. The concentration of **26c** decreased in plasma by factor of >10 after 60 min but we were delighted to observe that plasma concentration of C-terminal conjugates **20c** and **20h** at 60 min was indistinguishable from that of the parent **14c** macrocycles.

Encouraged by the initial evaluation of serum retention, we extended the study to 3 h. Gratifyingly we observed that the concentration of **14c** and SA-21 in plasma after 3 h were indistinguishable (Fig. 6C). Analogues of **14c** in which **PFS** was replaced by **HFB** (**14j**) or **DFB** (**14k**) perfluoroarenes consistently showed over 10x decrease in plasma concentration; a decrease of nearly 100x was observed for **14m** analogue in which perfluoroarene was substituted by metabromoxylene linker (**MBX**) (Fig. 6C). As expected, macrocycle **25c**, the F6A mutant of **14c**, was not detectable in serum at or after 60 min and the concentration of macrocycle **21c** S1A mutant of **14c** was significantly lower than **14c** at 3 h (Fig. 6D). After 3 h of circulation plasma concentrations of **PFS**-SICRFFCGGGZ (**20c**) at 3 h were indistinguishable from **14c** (Fig. 6E) whereas the **PFS**-SICRFFCGGK(N3) **26c** showed a suboptimal long-term performance in plasma (Supplementary Fig. 32). Similarly to the short-term observations (Fig. 6B), it appeared that the molecules with longer (Gly)$_3$-linker at the C-terminus provided better plasma retention than those with (Gly)$_2$ linker. We did not investigate the linker requirements in detail but we attached all subsequent payloads to the C-terminal end via (Gly)$_3$-linker.

The PK properties of the analogues that contained polyethylene glycol (PEG) moieties at N- or C-termini reinforced the above observation and identified the optimal location for attachment of therapeutically-relevant payloads such as peptide apelin analogues[53–56]. The macrocycle **14g** synthesized from **14c** by acylation of N-terminal Ser with azido-PEG$_6$-carboxylic acid was not detectable in plasma after 3 h whereas the concentration of C-terminal PEG-ylated compounds **20h** and **14i** after 3 h of circulation was indistinguishable from unmodified **PFS**-SICRFFCGGG. Compounds **14i** is a product of C-terminal amidation of **PFS**-SICRFFCGGG-COOH by NH$_2$-PEG$_5$-N$_3$, whereas in compound **20h** di-azido-PEG$_3$ linker was ligated to **PFS**-SICRFFCGGGZ (**20c**) by CuAAC. An indistinguishable plasma concentration of **14c**, **14i**, **20c**, and **20h** after 3 h suggested that structural changes after the (Gly)$_3$-linker are permissive for albumin recognition and do not affect long-term circulation in plasma. Gratifyingly we observed that conjugation of **PFS**-SICRFFCGGG to apelin-17 analogue (N$_3$-PEG$_6$-NMe17A) increased its plasma retention significantly. We could not detect the apelin-17 analogue in plasma even 2 min after injection and such rapid clearance of apelin analogues is consistent with previous observations[57]. In contrast, plasma concentration of compound **20n**, produced by CuAAC ligation of N$_3$-PEG$_6$-NMe17A[53] to **PFS**-SICRFFCGGGZ (**20c**) by Cu-catalysed 1,3-dipolar cycloaddition was indistinguishable from the plasma concentration of **20c** after 3 h. Such enhanced circulation of apelin-17 analogue promoted by a relatively small albumin-binding peptide is a promising starting point for more in-depth investigation and benchmarking to previously reported conjugates of apelin and albumin binding antibody[58] or apelin and 40 kDa PEG group[59].

## Discussion

Late-stage modification of peptides and genetically-encoded (GE) libraries of peptides by cross-linkers (linchpins) is one of the common approaches to incorporate beneficial attributes to their properties[60]. Alkylation of cysteine residues in peptides via an S$_N$2 reaction using bi- or tri-dentate alkyl halides has been used for cyclization of peptides, incorporation of unnatural fragments into the resulting macrocycles[47,61,62], and late-stage modification of phage- and mRNA-displayed libraries to yield billion-scale GE libraries. Peptide cyclization via S$_N$Ar reaction with perfluoroarenes popularized by the Pentelute group forms alkyl-aryl thioethers[45]; other classes reactions have been developed to form aryl[63–65], alkenyl and alkynyl thioethers[66] in unprotected peptides. Aryl and perfluoroaryl thioethers are more resistant toward oxidation when compared to traditional bis-alkyl thioesters[44]. Decreased conformational mobility or aryl-thioether bond has been proposed to equip the resulting macrocycles with favourable properties such as cell permeability and proteolytic stability[50,67]. Our report described the selection from perfluoro-aryl macrocyclic GE libraries. There exists only one example of GE selection from S$_N$Ar-modified phage libraries: Lu and co-workers recently employed 2,4-difluoro-6-hydroxy-1,3,5-benzenetricarbonitrile (DFB) as a reagent that can modify phage libraries in water[68]. Chen and co-workers also used Pd-catalysed S$_N$Ar reaction to yield DNA-encoded libraries[69]. Both S$_N$Ar reaction yield macrocycles do not contain any fluorine atoms. On the other hand, fluorine handles present in perfluoroaryl-cross-linked macrocycles offer a unique possibility to use of $^{19}$F NMR to measure protein-macrocycle interactions. Interaction of perfluorinated aryls with proteins is also electronically distinct from non-fluorinated aromatic residues and in some cases it can offer uniquely advantageous interactions[70].

An interesting observation in selection of GE **OFS**-macrocycle libraries is mild reactivity of these structures towards thiol nucleophiles[71]. Libraries of mild electrophiles[72–74] and phage-displayed libraries with built-in electrophiles[75] have emerged as an important starting point for discovery of covalent and reversibly covalent inhibitors. While we do not show it in our report, it is possible that an attenuated reactivity of **OFS**-macrocycles towards thiols can be used as an advantageous feature in discovery of inhibitors that form covalent bonds with thiol residues in proteins. If reactivity of the selected macrocycles is not desired, one can perform late-stage replacement of **OFS** moiety in the identified hits with nearly isosteric perfluorophenyl-sulfide. The **PFS** linchpin is not sufficiently reactive for direct modification of phage-displayed libraries in water, but replacement of **DFS** linchpin by **PFS** "post-discovery" maintains the conformation and binding affinity of the discovered macrocycles while alleviating their undesired electrophilicity. Our report, thus, suggests a general approach for the future utility of perfluoroaryl-modified libraries: Step1: select phage-displayed libraries of **OFS**-macrocycles against the desired target. Step 2: evaluate **PFS**-modified synthetic macrocycles for their ability to bind to these targets.

Human serum albumin target used in this publication is a commonly employed model target in screen of phage-displayed or DNA-encoded libraries (DEL) and traditional high-throughput screening (HTS). Albumin is a complex multi-pocket receptor with regions that can bind to fatty acid-like moieties, dicarboxylic acids as well a wide variety of aromatic and heterocyclic compounds and large dye molecules[51,76]. Albumin also contains several binding sites for peptides as well as small proteins that have been utilized for half-life extension strategies[23,77]. Small footprint peptide macrocycles discovered in this report add to a diverse set of known albumin binders. Small-molecules have been developed with astonishing affinity for human albumin, but such molecules exhibit no binding to mouse albumin and no retention in mice, unless such mouse is engineered to express human albumin[78]. Mid-size peptide macrocycle described in this report might provide an interesting opportunity for sufficient affinity but also much desired cross-reactivity between the species. The affinity of **PFS**-SICRFFCGGG (**14c**) (1–4 µM confirmed in FP and $^{19}$F NMR assays) is less than an order of magnitude away from the affinity of benchmark SA-21 (0.6 µM). An interesting role for (Gly)$_3$ linker at the C-terminus in PK studies suggests that a C-terminal augmentation to this scaffold is a productive avenue for fine-tuning of the affinity without increasing the size of the peptide. Remarkably, **14c** and its analogues with C-terminal extensions beyond (Gly)$_3$ linker exhibited virtually identical retention times, mirroring the

benchmark albumin binder SA-21. This compelling finding underscores their potential as highly promising tools for albumin-binding applications. The small size of such peptide-macrocycle families makes it trivial to make them by solid-phase synthesis or incorporate them a part of another sequence produced by solid phase synthesis. More importantly, the SICRFFC motif can be easily re-introduced into phage-displayed libraries to serve as a constant N-terminal albumin binding motif and giving rise to libraries with predictable circulation half-life. The compact nature of the macrocycle allows introducing this albumin-binding moiety to diverse display platforms within the initial stages of screening of genetically-encoded libraries, making it possible to perform *de-novo* discovery of peptide and macrocyclic modalities with predictable circulation lifetime in animal models.

## Methods

### Preparation of SXCX₃C phage-displayed library

The procedures have been adopted and modified as previously described in publications that produced the M13-displayed SXCXXXC library[46] and M13-SDB vector[79] and SXCX₄C and SXCX₅C libraries[80]. In short, the vector SB4 QFT*LHQ was digested with Kpn I HF (NEB Cat# R3142S) and Eag I HF (NEB Cat# R3505S). A primer/template pair consisting of primer 5′-AT GGC GCC CGG CCG AAC CTC CAC C-3′ and template 5′-CC CGG GTA CCT TTC TAT TCT CAC TCT TCT X TGT XXX TGT GGT GGA GGT TCG GCC GGG CGC TTG ATT-3′ with 'X' representing a trinucleotide formed by annealing. The primer/template was then extended using Klenow DNA polymerase (NEB) according to the manufacturer's instructions. The insert fragment was then digested with Kpn1 HF and Eag1 HF, gel purified, and ligated into the cut vector. The ligation products were then transformed into electrocompetent *E. coli* cells, and the transformants were grown overnight on *E. coli* TG1 to allow for phage production. Phage cultures were then centrifuged to remove cells and debris, and then the phage was precipitated by PEG precipitation (5% PEG 0.5 M NaCl). Other SDB vectors have been processed identically. We sequenced the naive libraries by Illumina sequencing, and the naive library of SXCX$_n$C ($n$ = 3–5) composition is publicly available at the following link: https://48hd.cloud/file/1470.

### Panning campaigns

Three panning campaigns were conducted to discover binders for Human Serum Albumin (HSA) (Sigma-Aldrich, Cat# A4327-1G) with Protein A (Sigma-Aldrich, cat# P6031-1MG), Concanavalin A (Sigma-Aldrich, Cat# C2010-100MG) or T4-Gp as the negative controls. The proteins were immobilized on polystyrene plates or magnetic beads and panned against **OFS**-macrocyclic libraries or unmodified libraries. The details on panning experiments are discussed in supplementary information sections 3–5 entitled "Panning Strategy". The analyses were performed by next-generation sequencing (NGS) of phage DNA similarly to previous publications[47,62] and the details are described in Supplementary Information section 9 "Illumina Sequencing [...]" and Section 31 "MATLAB Script for DE analysis".

### Preparation of Illumina sequencing samples

Similar to previous reports[47,62], phage eluted from the target was subjected to PCR amplification to append Illumina multiplexing barcodes, and sequencing adaptors to randomized library regions (see Supplementary Information section 7 entitled "7. PCR amplification protocol for Illumina deep sequencing"). All PCR products were quantified by 2% (w/v) agarose gel in Tris-Borate-EDTA buffer at 100 volts for ~35 min using a low molecular weight DNA ladder as standard (NEB, Cat# N3233S). PCR products that contain different indexing barcodes were pooled, allowing 10 ng of each product in the mixture. The mixture was purified by eGel, quantified by quBit and sequenced using the Illumina NextSeq paired-end 500/550 High Output Kit v2.5 (2 × 75 Cycles). Data were automatically uploaded to BaseSpace™ Sequence Hub.

### Processing of Illumina data

The Gzip compressed FASTQ files were downloaded from BaseSpace™ Sequence Hub. The files were converted into tables of DNA sequences and their counts per experiment. Briefly, FASTQ files were parsed based on unique multiplexing barcodes within the reads discarding any reads that contained a low-quality score. Mapping the forward (F) and reverse (R) barcoding regions, mapping of F and R priming regions allowing no more than one base substitution each and F-R read alignment allowing no mismatches between F and R reads yielded DNA sequences located between the priming regions as described in previous publications[62]. The files with DNA reads, raw counts, and mapped peptide modifications were uploaded to http://48hd.cloud/ server. Each experiment has a unique alphanumeric name and unique static URL in Supplementary Tables 1–3.

### General protocol for cyclization with decafluorodiphenylsulfone

Procedure was analogous to previously published methods[44,45]. In short, linear peptide (10 mM) was dissolved in 50% acetonitrile and Tris buffer (50 mM Tris-HCl, pH 8.5), then 2 equivalents of **DFS** in 50% acetonitrile and Tris buffer (50 mM Tris-HCl, pH 8.5) was added to the mixture. The mixture was vortexed for 30 s, incubated for 2 h at room temperature, purified by HPLC and further lyophilized to yield product.

### General protocol for cyclization with pentaflurophenyl-sulfide

Procedure was analogous to previously published methods[44,45]. In short, linear peptide (10 mM) was dissolved in 50 mM Tris in DMF, then 2 equivalents of **PFS** (Matrix Scientific, Cat# 009083) was added to the mixture. The mixture was vortexed for 30 s and allow to react for 1 h at RT. The reaction mixture was purified by HPLC and lyophilized to yield the product.

### In vivo pharmacokinetic experiment

All the procedures and experiments involving animals were carried out using a protocol approved by the Health Sciences Laboratory Animal Services (HSLAS), University of Alberta. The protocol was approved as per the Canadian Council on Animal Care (CCAC) guidelines. Approved protocol # AUP00002467. All mice (Strain: C57BL/6J) were maintained in pathogen-free conditions at the University of Alberta breeding facility. Housed on ventilated caging (Tecniplast); Temperature: 21 +/− 2 degrees Celcius; Humidity: 30–70% rH; Light Cycle: 12 L:12D. Caging: 1 cm of aspen chip, 1 cotton Nestlet, 1 Bed'r'nest, 1 polycarbonate tube; Water: ad lib, deionized UV filtered water; Food: ad lib, irradiated Lab Diet 50LD Rodent Chow (4.5% fat). Peptide mixtures of 100 μM were prepared in PBS. Mice were administered with 200 μL of the peptide mixture solution with tail veil injection. A series of 6 blood samples were collected at time points from 2 min up to 240 min (Fig. 6B: 2 min, 60 min; Fig. 6C–E: 2 min, 60 min, 180 min; Supplementary Fig. 32: 2 min, 5 min, 30 min, 60 min, 120 min, 240 min). Samples were collected in tubes that contained sodium citrate as an anticoagulant and then centrifuged at 5 min at 2000 × $g$ to collect the blood plasma. 10 μL of plasma portion were transferred into a tube containing 40 μL of 8:2 acetonitrile/water to precipitate proteins. The samples were centrifuged at 17,000 × $g$ for 10 min at 4 °C. Supernatants were then transferred to new tubes and subjected to analysis by LC-MS.

### Reporting summary

Further information on research design is available in the Nature Portfolio Reporting Summary linked to this article.

## Data availability

Supplementary information document contains Supplementary Figs. 1–99, Tables 1–10, synthetic methods and characterization

of compounds, details of phage display selection, next-generation sequencing and bioinformatics analysis, and all biochemical assays. PDB files produced by docking, data for the MATLAB script and the MATLAB scripts are available at Supplementary data.zip. The DNA reads, raw counts, and mapped peptide modifications are available publicly at http://48hd.cloud/ server and the links to individual files are listed in Supplementary Tables 1–3. The HSA molecular docking scripts are available at https://doi.org/10.5281/zenodo.8165764.

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

## Acknowledgements

This research was supported by a research contract from the Ferring Research Institute, Natural Sciences and Engineering Research Council of Canada (NSERC, RGPIN-2016-402511 to R.D.), NSERC Accelerator Supplement (to R.D.), and Canadian Institutes of Health Research grant (CIHR FRN 168961, to R.D.). This work was also supported by the National Institute of General Medical Sciences of the National Institutes of Health under award number R01GM124160 (PI: Y.-S.L.) Infrastructure support was provided by CFI New Leader Opportunity (to R.D.). We thank Dr.

Randy Whittal for assistance with LCMS, and Mark Miskolzie for assistance with $^{19}$F NMR kinetics.

## Author contributions

R.D. conceived, designed, and supervised the overall project. E.R.A.-P., S.K.-M. and J.W. performed chemical modifications on phage display libraries and selection against albumin under the supervision of R.D., J.J.D. and J.M.N. W.C.K.P. designed and supervised all 19F NMR assays, J.W., A.E., R.Q., S.E.K. and J.L. performed the 19F NMR binding assays. J.W. performed and analysed all fluorescence polarization assays. J.W. and E.R.A.-P. performed and analysed all isothermal titration calorimetry assays. J.W., A.E., S.K., R.Q., E.R.A.-P. and S.K-M. chemically synthesized and characterized all phage display-derived peptides conjugates and K.F. synthesized and characterized all apelin conjugates under supervision of J.C.V. M.M., R.D. and A.E. designed in vivo studies. S.S. performed in vivo mice studies and A.E. performed analysis of all samples from in vivo studies. P.K., L.V., J.M., Y.-S.L., H.K. and H.C. performed computational studies. R.D., J.W. and A.E. wrote the manuscript and edited the final manuscript. All authors approved the final manuscript.

## Competing interests

The authors declare the following competing financial interests. Ratmir Derda is the founder and CEO of 48Hour Discovery Inc. The remaining authors declare no competing interests.
