## [Peer Review file · Nature Communications]

REVIEWER COMMENTS

Reviewer #1 (Remarks to the Author):

This is a relatively thorough and well-written article that describes the development of a macrocyclic peptide that binds to albumin. The research has the potential to be helpful in extending the serum stability of peptides, as the identified peptide can be incorporated into other peptide libraries to identify compounds that bind to other targets. However, the manuscript stopped short without showing any major application. A critical application will be necessary to show the real potential impact of the identified macrocyclic peptide. A direct comparison with the conjugation with a long fatty acid for prolonging serum half lives of a peptide drug will be necessary as well. Therefore, a major revision with a demonstrated application is required. Additional minor concerns are provided below.

1. Can the authors comment on why an N-terminal serine library was used? This N-terminal serine shows no purpose.
2. This may just be an issue with formatting, but Figure 2 A and B are confusing and it is hard to distinguish the parts of the figure.
3. Can the authors comment on the % cyclization of the 3mer phages being cyclized with OFS? Are there any experimental details confirming that such a small library will cyclize?
4. Could the authors expand on their reasoning to only perform one round of selection for the third campaign? Is this related to the total phage input compared to the library size?
5. Can the authors expand on their analysis of the ITC experiments and give a possible explanation for why they did not show 1:1 binding?
6. To highlight the necessity of the PFS linkage, it would be beneficial to compare the affinity to the linear peptide and/or disulfide cyclized peptide to show that they either do not bind or have considerably lower affinity.

7. “The switching from DFS to PFS also exhibited a minimal effect on the binding of peptides 5b and 5c (Figure S20).” This statement should be expanded – in the figure it looks like there is higher signal decrease by PFS compared to DFS.

8. Can the authors highlight the importance of binding to a different site than other known albumin binders? Also, is the binding site for SA-21 known and does it bind to a different site than 5c?

9. In Figure 8, what is the unit for the y axis on the peptide retention graph? Is the n = 3 referring to biological or technical replicates? It may be beneficial to report the calculated serum half-lives of the peptides.

Reviewer #2 (Remarks to the Author):

The present work describes the selection of phage-encoded macrocyclic peptides, modified with cysteine-reactive perfluoroaromatic linkers, that are capable of binding human serum albumin (HSA). The best isolated perfluoroaryl-macrocyclic peptide, named 5c, binds HSA with K_d values of 4 – 6 μM . The macrocycle 5c cross-reacts with murine albumin and does not bind two unrelated control proteins. Docking studies suggested that the primary HSA binding site for perfluoroaryl-macrocyclic peptide 5c is the fatty acid binding site 1 (FA1). Next, the half-life of compound 5c was assessed in vivo. Overall, the manuscript yields “mixed feelings”. I like the approach; the topic covered in this article is likely to be of interest to a reasonable number of scientists working in this field and certainly worthy of investigation. The manuscript is concise and easy to read. I am a little less enthusiastic about the characterization of the identified binder. To be honest, I am struggling with this flaw, I would have had less hesitation if the authors had assessed and validated their technology against multiple targets or at least provided a more thoughtful characterization of the isolated HSA binders. I don’t know if this is a Nature Communication-level work, but this should be an editor level decision.

The novelty is the “Achille’s heel” of this manuscript as it looks like more an extension of previous works by the same group, which, one may say, “scoops the originality of the approach”. Indeed, the design, chemical modification, and next-generation sequencing of such libraries have been previously detailed in “Compositional Bias in Naive and Chemically-modified Phage-Displayed Libraries uncovered by Paired-

end Deep Sequencing, *Sci. Rep.* 2018, 8 (1), 1214” (reference 44) and in “Rapid biocompatible macrocyclization of peptides with decafluoro-diphenylsulfone, *Chem. Sci.* 2016, 7 (6), 3785-3790” (reference 42) by the same group. The current manuscript adds an extra step and describes the application of such libraries for the selection of PFS-modified synthetic macrocycles against HSA.

The methodology is detailed and clearly described. The technology, as described, appears to work well for HSA.

The conclusions drawn by the authors are not always convincing or supported by clear experimental data. I recommend the authors to provide a more balanced evaluation of their findings as well as of the challenges that still remain. The technology as described is proved to work for one only target so far.

Below are listed a few points where the clarity of the manuscript can be improved. They are meant to make this work stronger and more impactful.

Major revisions:

1. Page 5, line 86: “The main mechanism leading to the long half-life of albumin and antibody are similar: both proteins interact with FcRn on the surface of immune cells”. This sentence is partially incorrect. The authors might have mistaken FcRn with FcγRs. FcRn is primarily expressed on the surface of endothelial cells, the major site of IgG catabolism and recycling, and on the surface of interstitial macrophages. I suggest the authors to edit the sentence and eventually cite these two works:

- Latvala S. et al., Distribution of FcRn Across Species and Tissues. *Journal of Histochemistry and Cytochemistry*, 65: 321–33

(<https://doi.org/10.1369/0022155417705095>)

- Rudnik-Jansen I., et al. FcRn expression in cancer: Mechanistic basis and therapeutic opportunities. *Journal of Controlled Release*, 337: 248-57

(<https://doi.org/10.1016/j.jconrel.2021.07.007>)

2. Page 7 (line 127). The authors state “There is a need for development of other albumin binding peptide modalities that have lower molecular weight”. Why do we need lower molecular weight molecules? What would be the advantages of smaller peptides with low affinity versus slightly bigger peptides but with higher affinity? Why not use then available small molecules capable of binding albumin? The literature is plenty of small molecules (e.g., Evans blue, drugs, fatty acids, etc.) that have

been successfully used to prolong the half-life of peptides and small proteins such as insulin. I suggest that the authors carefully clarify this point that, at the end, is the aim behind all their work on finding smaller peptide macrocycles against HSA.

3. Page 7 (line 133). The authors state “We hypothesized that a perfluoroaromatic linchpin would serve as useful pharmacophore recognized by one of the binding sites of HSA similarly to the binding of fatty acid in lipidated peptides”. Do the authors have experimental evidence supporting their hypothesis that the two perfluoroaromatic compounds (OFS and PFS) applied in this study are capable of binding HSA? If yes, please provide the experimental data. If no, please measure the binding affinity of both OFS and PFS to HSA.

4. Page 9. The difference in enrichment observed between the first (2-fold increase) and the second (200-fold increase) discovery campaigns against HSA is impressive. Please provide a critical and thoughtful explanation for this.

5. Page 9 and 10. Different selection strategies employed against the same target yielded different peptide consensus sequences. Please provide a critical and thoughtful explanation for this.

6. Page 14. The fact that two macrocyclic peptides, 5c and 6c, that differ by only three amino acids, display a 100-fold difference in K_d values is remarkable and must be stressed. This data would support the fact that the binding is not driven by the perfluoroaromatic linker only, but the peptide also appears to play a role. Please provide additional insights on this finding.

7. I must admit that I am very surprised by the ITC data shown in Figure S13, S14 and S15. The K_D values reported for SA-21 peptide are very far from those described in literature (~15-fold higher in figure S13, ~100-fold higher in figure S14 and ~25-fold higher in figure S15). This makes me think that something in the design of the experiment, preparation of the protein and/or in the setting of the instrument went wrong. I suggest that the authors characterize macrocycle binding using a complementary technique such as SPR, which is the same technique used by Dennis and colleagues to determine the affinity of SA-21 ($K_d = 467$ nM). I must also add that the multi-site binding behaviour observed by the authors for all peptides when performing ITC measurements is not surprising as HSA is known to be highly flexible and undergoes conformational changes upon binding.

8. Page 16. Docking studies indicate that perfluoroaryl-macrocyclic 5c binds fatty acid binding site 1 (FA1). However, no experimental data have been shown in support of this. The authors should experimentally confirm their *in silico* findings. There are several commercially available drugs capable of binding FA1 site (please see two manuscripts listed below) that can be used as probe in competitive NMR, FP, ITC and SPR studies. Why not try to co-crystallise 5c in complex with HSA?

- Ferenc Zsila. Subdomain IB Is the Third Major Drug Binding Region of Human Serum Albumin: Toward the Three-Sites Model. *Mol. Pharmaceutics* 2013, 10, 5, 1668–1682

<https://doi.org/10.1021/mp400027q>

- Wang Z.M. et al., Structural studies of several clinically important oncology drugs in complex with human serum albumin. *Biochim Biophys Acta* 1830: 5356-5374

DOI: 10.1016/j.bbagen.2013.06.032

9. Figure 6. The authors showed that the binding of macrocycle 5c did not decrease in the presence of any of the three commercially available drugs tested. However, if we carefully analyse the structure of HSA in complex with diclofenac described by Zhang Y. et al. (*Chem Biol Drug Des* 86: 1178-1184; PDB: 4Z69), we realize that one molecule of diclofenac is bound to FA1 site in the IB subdomain. If the docking experiments predict that 5c binds to FA1, why did the authors not observe competition when performing the 19F NMR experiments in the presence of diclofenac? Please provide a critical and thoughtful explanation for this.

10. Page 19. The authors should determine and provide all the major pharmacokinetic parameters such as elimination half-life ($t_{1/2}$), maximal concentration (C_{max}), volume of distribution (V_d), clearance (CL), and so forth.

11. Ideally, I would have liked to see a better characterization of the binding mode of macrocycle 5c in complex with HSA. For instance, it would be nice to see if 5c competes with other HSA-binding peptides (e.g., DX-263, SA-21 and FITC labeled linear peptide). Similarly, it would be interesting to investigate the metabolic stability of 5c in comparison to previously reported HSA-binding peptides. I believe all these additional data would add significance and robustness to the whole story.

Minor revisions:

1 Figure 1ABC. Chemical structures of previously reported albumin-binding peptides DX-263 (A), SA-21 (B) and FITC labeled linear peptide (C) don't add something new to the work and should, in my opinion, be moved to the supplementary information section.

2 Figure 1D. This scheme is somehow repeated in Figure 2 (top). One of the two draws is sufficient to understand the technology.

3 Figure 5A. Unit of measure on label y-axis “(μm)” should be edited to “(μM)”.

4 Figure 5B. Please add here also the FP assay for BODIPY labelled 5c titrated against mouse serum albumin.

5 Figure 6A. Please label the different fatty acid binding sites (FAs) on the two HSA structures reported. This will be helping the reader to quickly locate the different FA sites discussed in the text.

6 Figure 7A. The resolution of this figure is very low and must be improved. I also advise the author to label the different FAs on the HSA structure.

7 Figure 8. The author should plot the pharmacokinetic data as a canonical standard curve rather than bar diagrams.

8 There are several typos in the text which need to be addressed. I suggest that the authors read again the paper carefully.

Reviewer #3 (Remarks to the Author):

The authors present a new linkage using decafluoro-diphenylsulfone which they implement in a genetically encoded library to discover potent albumin binder. Albumin is an interesting target to prolong circulation half live of small molecules and peptides.

The presented strategy follows the idea of bridging peptides via cysteine functionalization, a well reported strategy of the generation of (bi-)cyclic peptides.¹ The octafluoro-diphenylsulfone crosslinked macrocyclic peptides represent a novel class of bridged macrocycle. The linker is unstable towards nucleophilic cysteines and was replaced by a nearly isosteric perfluorophenyl-sulfide for investigations on identified binders after the phage display. The authors showed that the replacement didn't alter binding. Still this can be considered as obstacle, that other bridging strategies do not face. The authors state, that this attenuated reactivity towards thiols can be used advantageous, to covalently bound to cysteines in proteins. No further experiments were performed into this direction.

The authors present an optimized phage display screening against albumin and identify enriched sequences. This process is underlined with a lot of thorough data. They further validate those candidates

by ^{19}F -NMR using the fluorinated handle. A main advantage of the strategy, according to them, is the implementation of a fluorinated linker that allows the use of ^{19}F -NMR. Despite showing the occurrence of binding events, the signals obtained from the NMR analyses are unsuitable for drawing conclusions on binding strength of the selected samples.

Furthermore, a fluorescence polarization binding assay was used to determine KDs of selected hits. The best binder achieves a KD-value of 4-6 μM . It competes with fatty acids (binding affinities between 0.5 and 60 μM) but lags behind the top peptidic binders (15 nM² or 39 nM [H. A. P. Revets and C. Boutton, Peptides capable of binding to serum albumin and compounds, constructs and polypeptides comprising the same, WO2011095545, 2011]).³ Regarding the small genetic library and approach of proving a concept, this value can be considered good.

In the next step they use computational methods to compare binding pockets of the most potent binder (5c) with known albumin binders. In my opinion, this experiment does not follow the thread of the report. The binding site of the peptide is not determined, the representation of the data is inadequate and I am missing the outcome of this paragraph. (Probably the authors are self-aware of that, because this part is also missing in the conclusion.)

Finally, the authors test their compound in vivo by evaluating serum stability of their top binder (5c) in mouse. The data shows increased half live circulation compared to the negative control. This experiment is a nice complementation of their in vitro data and totally reasonable for the stage of this project but is indeed, as stated, a minimalistic starting point.

The authors use a sound methodology to proof their new concept. The right conclusions are drawn from the presented data. Nevertheless, I am not convinced that this study is suitable for publication in nature communications yet. I can't see any superiority of the novel linker compared to established strategies. The need for replacement within the identification of a binder in combination with missing significance of the presented ^{19}F -NMR data aren't convincing. In my opinion the computational data is meaningless in the presented context.

To make this an impactful report I would expect the authors to pick up one of the loose ends within the story:

- 1) The attenuated reactivity of the octafluoro-diphenylsulfone macrocycles for covalent linkage as advantageous feature
- 2) Getting profound knowledge about the binding of the identified motive to albumin by profound computational studies or crystal structures
- 3) Establishment of a ^{19}F -NMR methodology with conclusive output

1. Heinis, C.; Rutherford, T.; Freund, S.; Winter, G., Phage-encoded combinatorial chemical libraries based on bicyclic peptides. *Nat Chem Biol* 2009, 5 (7), 502-7.

2. Zorzi, A.; Middendorp, S. J.; Wilbs, J.; Deyle, K.; Heinis, C., Acylated heptapeptide binds albumin with high affinity and application as tag furnishes long-acting peptides. *Nat Commun* 2017, 8, 16092.

3. Zorzi, A.; Linciano, S.; Angelini, A., Non-covalent albumin-binding ligands for extending the circulating half-life of small biotherapeutics. *Medchemcomm* 2019, 10 (7), 1068-1081.

REVIEWER COMMENTS

Reviewer #1 (Remarks to the Author):

This is a relatively thorough and well-written article that describes the development of a macrocyclic peptide that binds to albumin. The research has the potential to be helpful in extending the serum stability of peptides, as the identified peptide can be incorporated into other peptide libraries to identify compounds that bind to other targets.

We thank the reviewer for this positive evaluation of the manuscript.

RI.1. However, the manuscript stopped short without showing any major application. A critical application will be necessary to show the real potential impact of the identified macrocyclic peptide.

New results demonstrate that RFF peptide (**14c**) can extend the circulation half-life of attached payloads (e.g. therapeutically relevant peptide apelin-17 analogue).

We have incorporated the following data and figure (Fig. 6) into the main text.

“...we observed that conjugation of PFS-SICRFFCGGG to apelin-17 analogue (N₃-PEG₆-NMe17A) increased its plasma retention significantly. We could not detect apelin-17 analogue in plasma even 2 minutes after injection and such rapid clearance of apelin analogues is consistent with previous observations.¹ In contrast, plasma concentration of compound **20n**, produced by ligation of N₃-PEG₆-NMe17A² to PFS-SICRFFCGGGZ (**20c**) by Cu-catalyzed 1,3-dipolar cycloaddition was indistinguishable from the plasma concentration of **20c** after 3 hours. Such enhanced circulation of apelin-17 analogue promoted by a relatively small albumin-binding peptide is a promising starting point for more in-depth investigation and benchmarking to previously reported conjugates of apelin and albumin binding antibody³ or apelin and 40 kDa PEG group.⁴”

RI.2. A direct comparison with the conjugation with a long fatty acid for prolonging serum half lives of a peptide drug will be necessary as well. Therefore, a major revision with a demonstrated application is required.

Careful comparison of the attachment of all major albumin binders (including SA-21,⁵ 4-(*p*-iodophenyl)butyric acid,⁶ AlbudAb^{TM7}, Alubinder1⁸, and lipids) would require significant additional experimentation, including prohibitively expensive *in vivo* pharmacokinetic studies. It is beyond the scope of our current manuscript. None of the previous publications describing the discovery of albumin-binding peptides perform exhaustive comparisons to other albumin-binding modalities.⁵⁻⁸ Such studies usually appear in follow-up publications. Requesting it here introduces an unfair double standard.

Additional minor concerns are provided below.

RI.3. Can the authors comment on why an N-terminal serine library was used? This N-terminal serine shows no purpose.

In this publication, we used libraries previously developed and characterized in our lab. Our lab is generally interested in attachment of payloads via N-terminal Ser; hence all phage libraries produced in our lab contain a constant N-terminal Serine. The nature of the N-terminal residue is not relevant to the selection, and constant N-terminal residues are almost always present in phage display vectors to avoid biases associated with processing of randomized N-terminal residue. The

¹⁹F NMR binding assay showed that the K_D value increased from 3.6 μ M to 12.1 μ M for the Ser→Ala mutant.

We added the following sentences to the manuscript to clarify these points:

“The SXCX₃C library has been used in our previous publications successfully using only one round of sequencing⁹⁻¹¹. It was sufficiently small (160,000 variants) to be fully covered by NGS and copy number of each member of the naïve library was sufficiently large to afford quantification of the selection by NGS and DE analysis.”

We observed that R4A or F5A changes in **14c** ablated the binding between the perfluoro-macrocycles and HSA. In contrast, the S1A and I2A changes resulted in smaller changes in K_D of 3- and 10-fold respectively. The alanine scans indicate that variation to any of the four discovered amino acids in SXCXXC sequence negatively affects the binding. Even serine, a constant residue in selection, becomes modestly important in albumin recognition as well.”

R1.4. This may just be an issue with formatting, but Figure 2 A and B are confusing and it is hard to distinguish the parts of the figure.

We amended the figure **2A** and **B** and added the following clarification to the legend:

“Panning with chemically modified phage libraries (A) Modification of phage-displayed SXCX₃C disulfide library by **DFS** to yield **OFS-SXCX₃C** library (B) The **OFS-SXCX₃C** library panned against a mixture of biotinylated HSA and His-tag expressed T4-GP in solution containing unlabelled milk proteins. Targets were captured separately with avidin beads and Ni-NTA beads affinity beads. In the negative control, **OFS-SXCX₃C** library was panned against biotinylated ConA and captured with avidin beads.”

R1.5. Can the authors comment on the % cyclization of the 3mer phages being cyclized with OFS? Are there any experimental details confirming that such a small library will cyclize?

The **OFS** linchpin utilized in this study was adapted from a previous study, where we confirmed that **OFS** could modify 85% of the displayed peptides.¹² We further validated the conversion of peptides of SXCX₃C library in this manuscript. (New Supplementary Fig. S3).

Supplementary Fig. S3. Quantification of modification of SXC₃C libraries by DFS. (A) M13-phage displayed disulfide library was reduced with TCEP and the exposed cysteine thiols were modified with biotin-peg iodoacetamide (BIA). The streptavidin capture (capture 1) of the biotinylated library reveals the percentage of available cysteines (B) M13-phage displayed thiol library was modified with DFS (C) Left over cysteines were monitored by a second biotinylation with BIA followed by a streptavidin capture (capture 2) (D) The difference between capture 1 and 2 reveals the percentage of library modified by DFS.

R1.6. Could the authors expand on their reasoning to only perform one round of selection for the third campaign? Is this related to the total phage input compared to the library size?

We added the following clarification to the results: “The SXC₃C library has been used in our previous publications successfully using only one round of sequencing⁹⁻¹¹. It was sufficiently small (160,000 variants) to be fully covered by NGS and copy number of each member of the naïve library was sufficiently large to afford quantification of the selection by NGS and DE analysis.”

R1.7. Can the authors expand on their analysis of the ITC experiments and give a possible explanation for why they did not show 1:1 binding?

We added the following clarification to the results: “... multi-step binding to albumin in ITC experiments has been observed in previous reports.¹³”

R1.8. To highlight the necessity of the PFS linkage, it would be beneficial to compare the affinity to the linear peptide and/or disulfide cyclized peptide to show that they either do not bind or have considerably lower affinity

We thank the reviewer for this great suggestion!

We compared the circulation time of **14c** lead macrocycle with linear version (**14l**) and RFF macrocycle modified with different linchpins (**14j**, **14k**, **14m**).

We added the following text to the main text.

“The concentration of SA-21 in plasma at 2 and 60 min decreased subtly from 4×10^4 to 3×10^4 ng/mL whereas concentration of macrocycle **14c** in plasma was 3 and 1.5×10^4 ng/mL at the same time points (Fig. 6B). In contrast, four derivatives of **14c** were excreted at 2 minutes or were not detectable at 60 minutes. The fast-clearing analogues were linear SICRFFCGGG with the cysteines alkylated by iodoacetamide (**14l**), the alanine mutants PFS-SICAFFCGGG (**23c**) and PFS-SACRFFCGGG (**21c**), and the control PFS-STCQGECGGG sequence (**17c**);”

and

“Replacing PFS by hexafluorobenzene (**HFB**, **14j**) and decafluorobiphenyl (**DFB**, **14k**) did not ablate retention in plasma completely (Fig. 6A); however, concentration of **HFB**- and **DFB**-modified macrocycles after 60 minutes were factor of 10 lower when compared to PFS-modified **14c** parent. These observations reinforced the important structural features of **14c**: both the amino acids sequence of **14c** and specific conformational constraint of this sequence imposed by PFS linker are critical for even a short-term retention in circulation.”

and

“Encouraged by the initial evaluation of serum retention, we extended the study to three hours. Gratifyingly we observed that the concentration of **14c** and SA-21 in plasma after 3 hours were indistinguishable (Fig. 6C). Analogs of **14c** in which PFS was replaced by **HFB** (**14j**) or **DFB** (**14k**) perfluoroarenes consistently showed over 10x decrease in plasma concentration; a decrease of nearly 100x was observed for **14m** analogue in which perfluoroarene was substituted by metabromoxylene linker (**MBX**) (Fig. 6C).”

R1.9. “The switching from DFS to PFS also exhibited a minimal effect on the binding of peptides 5b and 5c (Figure S20).” This statement should be expanded – in the figure it looks like there is higher signal decrease by PFS compared to DFS.

We added the following clarification to the result section: “The switching from **DFS** (**18f**) to **PFS** (**18d**) exhibited no difference in FP experiment (Supplementary Fig. S18A), but minor differences were observed by NMR (Supplementary Fig. S18B).”.

R1.10. Can the authors highlight the importance of binding to a different site than other known albumin binders? Also, is the binding site for SA-21 known and does it bind to a different site than 5c?

Although SA-21 peptide was discovered more than 20 years ago and used in multiple publications, the binding site for SA-21 was never determined. As of today, the team of Dennis and co-workers have not been able to map the site either by inhibition studies or any other methods (personal email communication with Mark Dennis).

R1.11. In Figure 8, what is the unit for the y axis on the peptide retention graph? Is the $n = 3$ referring to biological or technical replicates? It may be beneficial to report the calculated serum half-lives of the peptides.

We have amended this figure. Note that it is now moved to the supplementary information section (Supplementary Fig. S32).

Reviewer #2 (Remarks to the Author):

The present work describes the selection of phage-encoded macrocyclic peptides, modified with cysteine-reactive perfluoroaromatic linkers, that are capable of binding human serum albumin (HSA). The best isolated perfluoroaryl-macrocyclic peptide, named 5c, binds HSA with K_d values of 4 – 6 μ M. The macrocycle 5c cross-reacts with murine albumin and does not bind two unrelated control proteins. Docking studies suggested that the primary HSA binding site for perfluoroaryl-macrocyclic peptide 5c is the fatty acid binding site 1 (FA1). Next, the half-life of compound 5c was assessed in vivo. Overall, the manuscript yields “mixed feelings”. I like the approach; the topic covered in this article is likely to be of interest to a reasonable number of scientists working in this field and certainly worthy of investigation. The manuscript is concise and easy to read.

We thank the reviewer for their positive feedback.

R2.1. I am a little less enthusiastic about the characterization of the identified binder. To be honest, I am struggling with this flaw, I would have had less hesitation if the authors had assessed and validated their technology against multiple targets or at least provided a more thoughtful characterization of the isolated HSA binders. I don't know if this is a Nature Communication-level work, but this should be an editor level decision.

Replies to comments **R1.1, R1.2, R2.11, R2.12, R2.14**, new figures 3, 5, 6, and Supplementary Figures S21–S28, S29–S32 provide a more thorough characterization of HSA binders

*R2.2. The novelty is the “Achille’s heel” of this manuscript as it looks like more an extension of previous works by the same group, which, one may say, “scoops the originality of the approach”. Indeed, the design, chemical modification, and next-generation sequencing of such libraries have been previously detailed in “Compositional Bias in Naive and Chemically-modified Phage-Displayed Libraries uncovered by Paired-end Deep Sequencing, *Sci. Rep.* 2018, 8 (1), 1214” (reference 44) and in “Rapid biocompatible macrocyclization of peptides with decafluoro-diphenylsulfone, *Chem. Sci.* 2016, 7 (6), 3785-3790” (reference 42) by the same group. The current manuscript adds an extra step and describes the application of such libraries for the selection of PFS-modified synthetic macrocycles against HSA. The methodology is detailed and clearly described. The technology, as described, appears to work well for HSA.*

Our manuscript describes a previously unreported selection of perfluoro-aryl libraries (specifically three separate selection campaigns of various perfluoro-aryl cyclic architectures). The initial synthesis of such libraries is cited correctly, and they have been appropriately cited in our manuscript. Characterization of the disulfide precursor libraries by next-generation sequencing is also cited appropriately, although the purpose of the cited manuscript is very divergent from the current manuscript, and it aimed to characterize sequencing and expression biases rather than any selection. We believe disclosing prior research directions should not be a barrier to this publication. Prior publications cover very different areas that do not overlap with most topics in this publication.

R2.3. The conclusions drawn by the authors are not always convincing or supported by clear experimental data. I recommend the authors to provide a more balanced evaluation of their findings as well as of the challenges that still remain. The technology as described is proved to work for one only target so far.

We provide additional experimental data per specific request of all three reviewers with the exception of very onerous requests (e.g. comparison of all known albumin binders *in-vivo*.)

Below are listed a few points where the clarity of the manuscript can be improved. They are meant to make this work stronger and more impactful.

Major revisions:

R2.4. Page 5, line 86: “The main mechanism leading to the long half-life of albumin and antibody are similar: both proteins interact with FcRn on the surface of immune cells”. This sentence is partially incorrect. The authors might have mistaken FcRn with FcγRs. FcRn is primarily expressed on the surface of endothelial cells, the major site of IgG catabolism and recycling, and on the surface of interstitial macrophages. I suggest the authors to edit the sentence and eventually cite these two works:

- *Latvala S. et al., Distribution of FcRn Across Species and Tissues. Journal of Histochemistry and Cytochemistry, 65: 321–33 (<https://doi.org/10.1369/0022155417705095>)*
- *Rudnik-Jansen I., et al. FcRn expression in cancer: Mechanistic basis and therapeutic opportunities. Journal of Controlled Release, 337: 248-57 (<https://doi.org/10.1016/j.jconrel.2021.07.007>)*

We thank the reviewers, we amended sentence as: “The main mechanism leading to the long half-life of albumin and antibody are similar: both proteins interact with FcRns on the surface of endothelial cells.¹⁴⁻¹⁵” We have cited the mentioned work.

R2.5. Page 7 (line 127). The authors state “There is a need for development of other albumin binding peptide modalities that have lower molecular weight”. Why do we need lower molecular weight molecules? What would be the advantages of smaller peptides with low affinity versus slightly bigger peptides but with higher affinity? Why not use then available small molecules capable of binding albumin? The literature is plenty of small molecules (e.g., evans blue, drugs, fatty acids, etc.) that have been successfully used to prolong the half-life of peptides and small proteins such as insulin. I suggest that the authors carefully clarify this point that, at the end, is the aim behind all their work on finding smaller peptide macrocycles against HSA.

Our manuscript already contained a proposed application of the short peptide structures: “Similarly short peptides that bind to HSA could be used in tandem with therapeutic peptide or protein sequences to dial in predictable half-life for such therapeutics. Such short albumin-binding peptides could empower the development of many future therapeutic peptides because they could be built into any genetically encoded peptide library (e.g., displayed on phage, RNA, and other platforms) to give rise to billion-scale libraries with predictable in vivo half-life. However, short HSA-binding peptides are scarce.”

Few other discussion points have been added to the main text.

“Small-molecules have been developed with astonishing affinity for human albumin, but such molecules exhibit no binding to mouse albumin and no retention in mice, unless such mouse is engineered to express human albumin.⁸ Mid-size peptide macrocycle described in this report might provide an interesting opportunity for sufficient affinity but also much desired cross-reactivity between the species.”

and

“... the SICRFFC-motif can be easily re-introduced into phage-displayed libraries to serve as a constant N-terminal albumin binding motif and giving rise to libraries with predictable circulation half-life. The compact nature of the macrocycle allows introducing this albumin-binding moiety to diverse display platforms within the initial stages of screening of genetically-encoded libraries, making it possible to perform *de-novo* discovery of peptide and macrocyclic modalities with predictable circulation life-time in animal models.”

R2.6 Page 7 (line 133). The authors state “We hypothesized that a perfluoroaromatic linchpin would serve as useful pharmacophore recognized by one of the binding sites of HSA similarly to the binding of fatty acid in lipidated peptides”. Do the authors have experimental evidence supporting their hypothesis that the two perfluoroaromatic compounds (OFS and PFS) applied in this study are capable of binding HSA? If yes, please provide the experimental data. If no, please measure the binding affinity of both OFS and PFS to HSA.

Based on all data collected to date, it is appropriate to amend the statement as:

“We hypothesized that a perfluoroaromatic linchpin might serve as a useful pharmacophore and it might be recognized by one of the binding sites of HSA similar to the binding of fatty acid; however, as we observed in the NMR studies and late-stage pharmacokinetic evaluation of albumin-binding macrocycles, the primary function of perfluoro-linchpin is to constrain the discovered peptide macrocycle in a productive albumin-binding conformation. While perfluoroaromatic linchpin alone does not equip a random peptide with albumin-binding properties, the change in the shape of perfluoroaromatic linchpin is detrimental to the albumin-binding properties and *in vivo* circulation.”

R2.7. Page 9. The difference in enrichment observed between the first (2-fold increase) and the second (200-fold increase) discovery campaigns against HSA is impressive. Please provide a critical and thoughtful explanation for this.

We added the following clarification: “...the increase in round-to-round phage recovery was only modest. Despite depletion on protein A, the enriched population bound equally to HSA and protein A (Supplementary Fig. S1D). To mitigate these problems, the second discovery campaign increased the stringency and altered the immobilization of HSA between 96-well plate in rounds 1 and 3 and biotinylated HSA immobilized onto streptavidin beads in round 2 (Supplementary Fig. S2A). Such changes to selection stringency led to 200-fold increase in recovery in round 3 when compared to rounds 1 and 2.”

R2.8. Page 9 and 10. Different selection strategies employed against the same target yielded different peptide consensus sequences. Please provide a critical and thoughtful explanation for this.

We added the following clarification: “In summary, three selection strategies against HSA yielded divergent binding motifs. Such divergence of selection campaigns is not surprising because protein immobilization, depletion and amplification strategies were different among these strategies selection.”

R2.9. Page 14. The fact that two macrocyclic peptides, 5c and 6c, that differ by only three amino acids, display a 100-fold difference in K_d values is remarkable and must be stressed. This data would support the fact that the binding is not driven by the perfluoroaromatic linker only, but the peptide also appears

to play a role. Please provide additional insights on this finding.

We added the following new text and data: “A substantial change in measured K_D values resulting from subtle amino-acid changes in PFS-macrocycles highlighted that the binding is not driven by the perfluoroaromatic linker alone. The peptide sequence plays a major role. Alanine scan of **14c** further reinforced this observation (Fig. 3B–3C). We observed that R4A or F5A changes in **14c** ablated the binding between the perfluoro-macrocycles and HSA. In contrast, the S1A and I2A changes resulted in smaller changes in K_D of 3- and 10-fold respectively. The alanine scans indicate that variation to any of the four discovered amino acids in SXCXXXC sequence negatively affects the binding. Even serine, a constant residue in selection, becomes modestly important in albumin recognition as well.”

R2.10. I must admit that I am very surprised by the ITC data shown in Figure S13, S14 and S15. The K_D values reported for SA-21 peptide are very far from those described in literature (~15-fold higher in figure S13, ~100-fold higher in figure S14 and ~25-fold higher in figure S15). This makes me think that something in the design of the experiment, preparation of the protein and/or in the setting of the instrument went wrong. I suggest that the authors characterize macrocycle binding using a complementary technique such as SPR, which is the same technique used by Dennis and colleagues to determine the affinity of SA-21 ($K_d = 467$ nM). I must also add that the multi-site binding behaviour observed by the authors for all peptides when performing ITC measurements is not surprising as HSA is known to be highly flexible and undergoes conformational changes upon binding.

We have struggled with ITC characterization of albumin-peptide interactions, and, as reviewers point out, we saw very unsatisfactory, variable outcomes. We agree that such a problem might be inherent to albumin peptide interactions, variability of albumin of preparation of albumin solutions, or other factors such as the inherent binding of albumin to nearly every physical surface. We note similar multi-site binding in other publications and a complete lack of ITC data, including the three cornerstone peptide binding papers from Genentech⁵, Dyax corp¹⁶, and the Heinis¹⁷ group. Our group attempted the SPR several times, but we observed a very unusual negative index of refraction change, in the interest of clarity, it might be best not to show these SPR data in this manuscript.

R2.11. Page 16. Docking studies indicate that perfluoroaryl-macrocycle 5c binds fatty acid binding site 1 (FA1). However, no experimental data have been shown in support of this. The authors should experimentally confirm their in silico findings. There are several commercially available drugs capable of binding FA1 site (please see two manuscripts listed below) that can be used as probe in competitive NMR, FP, ITC and SPR studies. Why not try to co-crystallise 5c in complex with HSA?

• Ferenc Zsila. Subdomain IB Is the Third Major Drug Binding Region of Human Serum Albumin: Toward the Three-Sites Model. *Mol. Pharmaceutics* 2013, 10, 5, 1668–1682

<https://doi.org/10.1021/mp400027q>

• Wang Z.M. et al., Structural studies of several clinically important oncology drugs in complex with human serum albumin. *Biochim Biophys Acta* 1830: 5356-5374
DOI: 10.1016/j.bbagen.2013.06.032

Re: Why not co-crystallize? This is an unusually onerous request for a review process. We would like to disclose that we tried crystallization for many years in collaboration with Scott Lovell, who is the director of the Protein Structure and X-Ray Crystallography Laboratory at the University of Kansas. Our attempts to co-crystallize **14c** with HSA failed. Given the uncertainty of crystallization and prohibitive expenses, we discontinued our attempts.

The following text was added to main text

“To identify the binding location of PFS-SICRFFCGGG (**14c**), we attempted to co-crystallize **14c** with HSA but were not successful.”

The following text was added to the main text to provide our best attempts to link computational and experimental predictions:

“Nine distinct sites on HSA bind to fatty acids¹⁸, some of which also accommodate ibuprofen and diclofenac¹⁹⁻²⁰ (Supplementary Fig. S21, Fig. 5A). Docking of **14c** to these nine sites identified binding site 1, which is close to Hemin binding site, as the most favorable (Fig. 5A). The docking score for site 1 averaged over all the docking calculations across five distinct HSA structures (PDB 1e7e, 1e7f, 1e7g, 1e7h, 1e7i) was -8.95 ± 1.0 kcal/mol whereas the scores for the next sites were -6.6 ± 1.1 , -6.2 ± 0.7 , and -6.0 ± 1.2 kcal/mol (Supplementary Fig. S22). Observed docking preference away from known ibuprofen and diclofenac sites corroborated the experimental observation (Supplementary Fig. S22 and S23). As site 1 is hemin-binding site, a desired experimental validation of this prediction would be inhibition of HSA:**14a** interaction by hemin. We attempted to measure **14c**:HSA interactions in the presence of hemin, but the results were inconclusive due to strong association of hemin and **14c** in ¹⁹F NMR experiments.”

and

“The PMF corroborated that pocket 1 has significantly stronger binding energy than the others (Fig. 5A, Supplementary Fig. S24). Furthermore, the calculated $\Delta\Delta G$ of free energy perturbation calculations (FEP) for **14c** and five Ala mutants of **14c** were in alignment with K_D measured for the same alanine mutants in the ¹⁹F NMR assay (Fig. 3B–C). The biggest loss of function for R4A mutant, highlighted the importance of Arginine interaction for albumin binding (Fig. 5E–G); in contrast, both NMR and FEP agreed on a relatively minor role of N-terminal Ser and minor loss for S1A mutant.”

R2.12. Figure 6. The authors showed that the binding of macrocycle 5c did not decrease in the presence of any of the three commercially available drugs tested. However, if we carefully analyse the structure of HSA in complex with diclofenac described by Zhang Y. et al. (Chem Biol Drug Des 86: 1178-1184; PDB: 4Z69), we realize that one molecule of diclofenac is bound to FA1 site in the IB subdomain. If the docking experiments predict that 5c binds to FA1, why did the authors not observe competition when performing the 19F NMR experiments in the presence of diclofenac? Please provide a critical and thoughtful explanation for this.

We added the following explanation to the supplementary information section.

“When examining the superposition of diclofenac with **14c** (docking) in IB site of HAS (Supplementary Fig. S21), the overlap of **14c** and diclofenac occurs only on one side of **14c**. **14c** binds to 19 amino acids of HSA, and diclofenac binds to 9 amino acids of HSA, and 7 out of 9 amino acids which bind to diclofenac also bind to **14c** (L154, G189, Y161, K190, R186, H146, I142). Furthermore, the IB site of HSA in the crystal structure 4z69 binds simultaneously to both diclofenac and a fatty acid so that diclofenac binding could be modulated by the fatty acid binding. There is no fatty acid in the **14c**-HSA complex obtained by docking. Therefore, **14c** has a larger binding site to FA1 than diclofenac, and a more favorable docking score of -10 kcal/mol, compared to diclofenac’s binding score of -6.6 kcal/mol (Supplementary Fig. S21D).”

R2.13. Page 19. The authors should determine and provide all the major pharmacokinetic parameters such as elimination half-life ($t_{1/2}$), maximal concentration (C_{max}), volume of distribution (V_d),

clearance (CL), and so forth.

We have expanded the pharmacokinetic study section extensively and characterized multiple analogs of the lead binder (see new Fig.6 and accompanying discussion), but due to limited financial resources of our academic research group, when compared to other publications driven by well-funded pre-clinical divisions of major pharmaceutical companies, we have no financial means to run a full scale pre-clinical pharmacology of all compounds.

R2.14. Ideally, I would have liked to see a better characterization of the binding mode of macrocycle 5c in complex with HSA. For instance, it would be nice to see if 5c competes with other HSA-binding peptides (e.g., DX-236, SA-21 and FITC labeled linear peptide). Similarly, it would be interesting to investigate the metabolic stability of 5c in comparison to previously reported HSA-binding peptides. I believe all these additional data would add significance and robustness to the whole story.

Please see the responses for *R2.11*, *R1.1*, and *R1.2*.

R2.15. Figure 1ABC. Chemical structures of previously reported albumin-binding peptides DX-263 (A), SA-21 (B) and FITC labeled linear peptide (C) don't add something new to the work and should, in my opinion, be moved to the supplementary information section.

We believe that Figure 1 provides a clear summary of the existing peptide-based Albumin binders.

R2.16. Figure 1D. This scheme is somehow repeated in Figure 2 (top). One of the two draws is sufficient to understand the technology.

We thank the reviewer for pointing this out. The purpose of Fig. 1 is to compare the ligand introduced in this work to previous ligands. The purpose of Fig.2 is to summarize the chemical modification in the screening campaign.

R2.17. Figure 5A. Unit of measure on label y-axis “(μm)” should be edited to “(μM)”.

We amended the unit on the y axis (now, Fig. 4A)

R2.18. Figure 5B. Please add here also the FP assay for BODIPY labelled 5c titrated against mouse serum albumin.

We have incorporated the titration against mouse serum albumin (now, Fig. 4B)

R2.19. Figure 6A. Please label the different fatty acid binding sites (FAs) on the two HSA structures reported. This will be helping the reader to quickly locate the different FA sites discussed in the text.

We have incorporated a figure that clearly shows fatty acid binding sites (now, Fig. 5A).

R2.20. Figure 7A. The resolution of this figure is very low and must be improved. I also advise the author to label the different FAs on the HSA structure.

We have replaced the figure with a better-quality version and moved this figure to the supplementary information section. (now, Supplementary Fig. S32)

R2.21. Figure 8. The author should plot the pharmacokinetic data as a canonical standard curve rather than bar diagrams.

We have extended the data and discussion on pharmacokinetic properties of **14c** and its analogues. New Figure 6 is presented as curves for ng/mL of compounds.

R2.22. There are several typos in the text which need to be addressed. I suggest that the authors read again the paper carefully.

We thank the reviewer for their feedback, and we have corrected the typos to the best of our ability.

Reviewer #3 (Remarks to the Author):

R3.1. *The authors present a new linkage using decafluoro-diphenylsulfone which they implement in a genetically encoded library to discover potent albumin binder. Albumin is an interesting target to prolong circulation half live of small molecules and peptides. The presented strategy follows the idea of bridging peptides via cysteine functionalization, a well reported strategy of the generation of (bi-)cyclic peptides.¹ REF: I. Heinis, C.; Rutherford, T.; Freund, S.; Winter, G., Phage-encoded combinatorial chemical libraries based on bicyclic peptides. Nat Chem Biol 2009, 5 (7), 502-7.*

We thank the authors for the evaluation and historical context.

R3.2. *The octafluoro-diphenylsulfone crosslinked macrocyclic peptides represent a novel class of bridged macrocycle. The linker is unstable towards nucleophilic cysteines and was replaced by a nearly isosteric perfluorophenyl-sulfide for investigations on identified binders after the phage display. The authors showed that the replacement didn't alter binding. Still this can be considered as obstacle, that other bridging strategies do not face.*

We thank the reviewer for their evaluation. While the unwanted reactivity of **OFS** was an obstacle during the characterization stage, we overcame it by replacement of **DFS** with **PFS**.

R3.3. *The authors state, that this attenuated reactivity towards thiols can be used advantageous, to covalently bound to cysteines in proteins. No further experiments were performed into this direction.*

We believe this type of experiment is beyond the scope of this manuscript and hope to explore this avenue in future projects. It would not be fair to diminish the value of the current manuscript based on the forward-looking proposals in the conclusion section.

R3.4. *The authors present an optimized phage display screening against albumin and identify enriched sequences. This process is underlined with a lot of thorough data. They further validate those candidates by ¹⁹F-NMR using the fluorinated handle. A main advantage of the strategy, according to them, is the implementation of a fluorinated linker that allows the use of ¹⁹F-NMR. Despite showing the occurrence of binding events, the signals obtained from the NMR analyses are unsuitable for drawing conclusions on binding strength of the selected samples.*

We appreciate the reviewer's comments on our use of ¹⁹F NMR for characterizing the binding of our lead candidate. We believe that the reviewer underestimates the power of ¹⁹F NMR, as it has been shown to be a valuable tool for studying ligand-protein interactions in many previous studies. We would like to clarify that our use of ¹⁹F NMR allowed us to determine the K_D value for binding and observe binding events, which are crucial pieces of information for understanding the mechanism of action of our lead candidate.

Furthermore, we obtained quantitative K_D and structure-activity relationship data for alanine mutants of the lead candidate using ¹⁹F NMR.

To illustrate the sheer beauty and power of ¹⁹F NMR incorporate the ¹⁹F NMR measurements in main text Fig. 3, and include this figure here for the reviewer's convenience.

Fig. 3: ¹⁹F NMR measurement of macrocycle-albumin interactions. (A) ¹⁹F NMR binding assay for macrocycles **12c** (PFS-STCHANCGKKK-DFS) **14c** (PFS-SICRFFCGGG), **15c** (PFS-SFCPMFCGGG), **16c** (PFS-SLCKRECGGG), and **17c** (PFS-STCQGECEGGG) at 20 μM with against varying concentrations of HSA. (B) ¹⁹F NMR signals (C) and extrapolated binding curves for **14c** (PFS-SICRFFCGGG) and five alanine mutants (**21c**, **22c**, **23c**, **24c**, **25c**) of this macrocycle.

R3.5. Furthermore, a fluorescence polarization binding assay was used to determine K_D s of selected hits. The best binder achieves a K_D -value of 4-6 μM. It competes with fatty acids (binding affinities between 0.5 and 60 μM) but lags behind the top peptidic binders (15 nM² or 39 nM [H. A. P. Revets and C. Boutton, Peptides capable of binding to serum albumin and compounds, constructs and polypeptides comprising the same, WO2011095545, 2011]).³ Regarding the small genetic library and approach of proving a concept, this value can be considered good.

REF

2. Zorzi, A.; Middendorp, S. J.; Wilbs, J.; Deyle, K.; Heinis, C., Acylated heptapeptide binds albumin with high affinity and application as tag furnishes long-acting peptides. Nat Commun 2017, 8, 16092.

3. Zorzi, A.; Linciano, S.; Angelini, A., *Non-covalent albumin-binding ligands for extending the circulating half-life of small biotherapeutics. Medchemcomm* 2019, 10 (7), 1068-1081.

We thank the reviewer for this analysis and designation of the identified ligands as “good“

R3.6. In the next step they use computational methods to compare binding pockets of the most potent binder (5c) with known albumin binders. In my opinion, this experiment does not follow the thread of the report. The binding site of the peptide is not determined, the representation of the data is inadequate and I am missing the outcome of this paragraph. (Probably the authors are self-aware of that, because this part is also missing in the conclusion.)

We added the following summary:

“In summary, while we failed to co-crystallize **14c** with HSA, the combined findings from FP and NMR, drug inhibition, docking/MD, and aligned performance of Ala-mutants in FEP and NMR studies offer a useful guide for optimization of **14c** and its use in delivery. For example, both the binding pose in site1, and NMR/FP studies suggest that both C- and N- termini of the **14c** might be accessible as a plausible location for the attachments of payloads to **14c**. We followed up on these suggestions in pharmacokinetic studies.”

We also expanded the computational studies and their comparison to experimental as summarized below:

“To further evaluate prioritization of albumin site 1 as a plausible binding site for **14c**, we performed NVT (constant temperature, constant volume) and NPT (constant temperature, constant pressure) simulations of **14c** in site 1 of HSA solvated with explicit TIP3P water molecules and calculated binding free energy using steered molecule dynamics (SMD) and umbrella sampling techniques. We then calculated the potential of mean force (PMF) of the unbinding process in biased MD simulations with a harmonic potential whose interaction center is located at a specific distance between the binding pocket and the center of mass of **14c**. The ΔG was about -7.0 kcal/mol for pocket 1, while the ΔG for the other pockets was less than -4.2 kcal/mol. The PMF corroborated that pocket 1 has significantly stronger binding energy than the others (Fig. 5A, Supplementary Fig. S26). Furthermore, the calculated $\Delta\Delta G$ of free energy perturbation calculations (FEP) for **14c** and five Ala mutants of **14c** were in alignment with K_D measured for the same alanine mutants in the ^{19}F NMR assay (Fig. 3B–C). The biggest loss of function for R4A mutant highlighted the importance of Arginine interaction for albumin binding (Fig. 5E–G); in contrast, both NMR and FEP agreed on a relatively minor role of N-terminal Ser and minor loss for S1A mutant.”

R3.7. Finally, the authors test their compound in vivo by evaluating serum stability of their top binder (5c) in mouse. The data shows increased half live circulation compared to the negative control. This experiment is a nice complementation of their in vitro data and totally reasonable for the stage of this project but is indeed, as stated, a minimalistic starting point.

We invite the reviewer to see new *in vivo* experiments summarized in Fig.6 and answers to **RI.1** and **RI.2**

R3.8. The authors use a sound methodology to proof their new concept. The right conclusions are drawn from the presented data. Nevertheless, I am not convinced that this study is suitable for publication in nature communications yet. I can't see any superiority of the novel linker compared to established strategies. The need for replacement within the identification of a binder in combination

with missing significance of the presented 19F-NMR data aren't convincing. In my opinion the computational data is meaningless in the presented context.

We are confident that new data added to this manuscript (Fig. 3, 5, and 6) address these concerns.

R3.9. To make this an impactful report I would expect the authors to pick up one of the loose ends within the story:

- 1) The attenuated reactivity of the octafluoro-diphenylsulfone macrocycles for covalent linkage as advantageous feature
- 2) Getting profound knowledge about the binding of the identified motive to albumin by profound computational studies or crystal structures
- 3) Establishment of a 19F-NMR methodology with conclusive output

Answers to **R3.6** and **R2.11** expand on our docking MD simulations and add desired (profound) knowledge. For example, alanine mutants and KD results obtained by 19F NMR are well-correlated. These observations are further corroborated by *in vivo* pharmacokinetic studies (Fig. 6).

References

Please note that the numbers of the references here are different from the numbers used in the main text.

1. Japp, A. G.; Cruden, N. L.; Barnes, G.; van Gemeren, N.; Mathews, J.; Adamson, J.; Johnston, N. R.; Denvir, M. A.; Megson, I. L.; Flapan, A. D.; Newby, D. E., Acute Cardiovascular Effects of Apelin in Humans Potential Role in Patients With Chronic Heart Failure. *Circulation* **2010**, *121* (16), 1818-U56.
2. Fernandez, K. X.; Fischer, C.; Vu, J.; Gheblawi, M.; Wang, W.; Gottschalk, S.; Iturrioz, X.; Llorens-Cortes, C.; Oudit, G. Y.; Vederas, J. C., Metabolically stable apelin-analogues, incorporating cyclohexylalanine and homoarginine, as potent apelin receptor activators. *RSC Med. Chem.* **2021**, *12* (8), 1402-1413.
3. Read, C.; Yang, P.; Kuc, R. E.; Nyimanu, D.; Williams, T. L.; Glen, R. C.; Holt, L. J.; Arulanantham, H.; Smart, A.; Davenport, A. P.; Maguire, J. J., Apelin peptides linked to anti-serum albumin domain antibodies retain affinity in vitro and are efficacious receptor agonists in vivo. *Basic Clin. Pharmacol. Toxicol.* **2020**, *126 Suppl 6*, 96-103.
4. Jia, Z. Q.; Hou, L.; Leger, A.; Wu, I.; Kudej, A. B.; Stefano, J.; Jiang, C.; Pan, C. Q.; Akita, G. Y., Cardiovascular effects of a PEGylated apelin. *Peptides* **2012**, *38* (1), 181-8.
5. Dennis, M. S.; Zhang, M.; Meng, Y. G.; Kadkhodayan, M.; Kirchhofer, D.; Combs, D.; Damico, L. A., Albumin binding as a general strategy for improving the pharmacokinetics of proteins. *J. biol. chem.* **2002**, *277* (38), 35035-43.
6. Dumelin, C. E.; Trussel, S.; Buller, F.; Trachsel, E.; Bootz, F.; Zhang, Y.; Mannocci, L.; Beck, S. C.; Drumea-Mirancea, M.; Seeliger, M. W.; Baltes, C.; Muggler, T.; Kranz, F.; Rudin, M.; Melkko, S.; Scheuermann, J.; Neri, D., A portable albumin binder from a DNA-encoded chemical library. *Angew. Chem.* **2008**, *47* (17), 3196-3201.
7. O'Connor-Semmes, R. L.; Lin, J.; Hodge, R. J.; Andrews, S.; Chism, J.; Choudhury, A.; Nunez, D. J., GSK2374697, a novel albumin-binding domain antibody (AlbudAb), extends systemic exposure of exendin-4: first study in humans--PK/PD and safety. *Clin. Pharmacol. Ther.* **2014**, *96* (6), 704-12.
8. Vantourout, J. C.; Mason, A. M.; Yuen, J.; Simpson, G. L.; Evindar, G.; Kuai, L.; Hobbs, M.; Edgar, E.; Needle, S.; Bai, X.; Wilson, S.; Scott-Stevens, P.; Traylen, W.; Lambert, K.; Young, N.; Bunally, S.; Summerfield, S. G.; Snell, R.; Lad, R.; Shi, E.; Skinner, S.; Shewchuk, L.; Watson, A. J. B.; Chung, C. W.; Pal, S.; Holt, D. A.; Kallander, L. S.; Prendergast, J.; Rivera, K.; Washburn, D. G.;

- Harpel, M. R.; Arico-Muendel, C.; Isidro-Llobet, A., In Vivo Half-Life Extension of BMP1/TLL Metalloproteinase Inhibitors Using Small-Molecule Human Serum Albumin Binders. *Bioconjug. Chem.* **2021**, *32* (2), 279-289.
9. Ekanayake, A. I.; Sobze, L.; Kelich, P.; Youk, J.; Bennett, N. J.; Mukherjee, R.; Bhardwaj, A.; Wuest, F.; Vukovic, L.; Derda, R., Genetically Encoded Fragment-Based Discovery from Phage-Displayed Macrocyclic Libraries with Genetically Encoded Unnatural Pharmacophores. *J. Am. Chem. Soc.* **2021**, *143* (14), 5497-5507.
10. Triana, V.; Derda, R., Tandem Wittig/Diels-Alder diversification of genetically encoded peptide libraries. *Org. Biomol. Chem.* **2017**, *15* (37), 7869-7877.
11. He, B.; Tjhung, K. F.; Bennett, N. J.; Chou, Y.; Rau, A.; Huang, J.; Derda, R., Compositional Bias in Naive and Chemically-modified Phage-Displayed Libraries uncovered by Paired-end Deep Sequencing. *Sci. Rep.* **2018**, *8* (1), 1214.
12. Kalhor-Monfared, S.; Jafari, M. R.; Patterson, J. T.; Kitov, P. I.; Dwyer, J. J.; Nuss, J. M.; Derda, R., Rapid biocompatible macrocyclization of peptides with decafluoro-diphenylsulfone. *Chem. Sci.* **2016**, *7* (6), 3785-3790.
13. Callmann, C. E.; LeGuyader, C. L. M.; Burton, S. T.; Thompson, M. P.; Hennis, R.; Barback, C.; Henriksen, N. M.; Chan, W. C.; Jaremko, M. J.; Yang, J.; Garcia, A.; Burkart, M. D.; Gilson, M. K.; Momper, J. D.; Bertin, P. A.; Gianneschi, N. C., Antitumor Activity of 1,18-Octadecanedioic Acid-Paclitaxel Complexed with Human Serum Albumin. *J. Am. Chem. Soc.* **2019**, *141* (30), 11765-11769.
14. Latvala, S.; Jacobsen, B.; Otteneder, M. B.; Herrmann, A.; Kronenberg, S., Distribution of FcRn Across Species and Tissues. *J. Histochem. Cytochem.* **2017**, *65* (6), 321-333.
15. Rudnik-Jansen, I.; Howard, K. A., FcRn expression in cancer: Mechanistic basis and therapeutic opportunities. *J. Control. Release.* **2021**, *337*, 248-257.
16. Sato, A. K.; Sexton, D. J.; Morganelli, L. A.; Cohen, E. H.; Wu, Q. L.; Conley, G. P.; Streltsova, Z.; Lee, S. W.; Devlin, M.; DeOliveira, D. B.; Enright, J.; Kent, R. B.; Wescott, C. R.; Ransohoff, T. C.; Ley, A. C.; Ladner, R. C., Development of Mammalian Serum Albumin Affinity Purification Media by Peptide Phage Display. *Biotechnol. Prog.* **2002**, *18* (2), 182-192.
17. Zorzi, A.; Middendorp, S. J.; Wilbs, J.; Deyle, K.; Heinis, C., Acylated heptapeptide binds albumin with high affinity and application as tag furnishes long-acting peptides. *Nat. Commun.* **2017**, *8*, 16092.
18. Bhattacharya, A. A.; Grune, T.; Curry, S., Crystallographic analysis reveals common modes of binding of medium and long-chain fatty acids to human serum albumin. *J. Mol. Biol.* **2000**, *303* (5), 721-32.
19. Ghuman, J.; Zunszain, P. A.; Petitpas, I.; Bhattacharya, A. A.; Otagiri, M.; Curry, S., Structural basis of the drug-binding specificity of human serum albumin. *J. Mol. Biol.* **2005**, *353* (1), 38-52.
20. Zhang, Y.; Lee, P.; Liang, S.; Zhou, Z.; Wu, X.; Yang, F.; Liang, H., Structural basis of non-steroidal anti-inflammatory drug diclofenac binding to human serum albumin. *Chem. Biol. Drug Des.* **2015**, *86* (5), 1178-84.

REVIEWERS' COMMENTS

Reviewer #1 (Remarks to the Author):

The authors have addressed most concerns from this reviewer. The manuscript in the current form is ready for acceptance.

Reviewer #2 (Remarks to the Author):

The authors have properly addressed most of my comments and edited the identified inaccuracies. I therefore recommend to accept this manuscript for publication.

Reviewer #3 (Remarks to the Author):

The authors significantly elevated the paper by addressing the reviewers concerns.

The additional data for the ^{19}F NMR experiment and the improved visualization in the main text (Fig.3) highlights the strengths of this method better than in the manuscript submitted before.

The complementation of docking data as well as the rewriting of this paragraph made this part of the story more cohesive with the main story.

The major improvement at this point was the new *in vivo* data obtained. The identification of a potential ligation site for a therapeutic relevant cargo and the demonstration of prolonged serum stability brings this considerably towards an application.

In my opinion this paper can be accepted.